# FAST AND NOISE-ROBUST DIFFUSION SOLVERS FOR INVERSE PROBLEMS: A FREQUENTIST APPROACH

## ABSTRACT

Diffusion models have been firmly established as principled zero-shot solvers for linear and nonlinear inverse problems, owing to their powerful image prior and ease of formulation as Bayesian posterior samplers. However, many existing solvers struggle in the noisy measurement regime, either overfitting or underfitting to the measurement constraint, resulting in poor sample quality and inconsistent performance across noise levels. Moreover, existing solvers rely on approximating $x_0$ via Tweedie's formula, where an intractable *conditional* score is replaced by an *unconditional* score network, introducing a fundamental source of error in the resulting solution. In this work, we propose a novel frequentist's approach to diffusion-based inverse solvers, where each diffusion step can be seen as the maximum likelihood solution to a simple single-parameter conditional likelihood model, derived by an adjusted application of Tweedie's formula to the forward measurement model. We demonstrate that this perspective is not only scalable and fast, but also allows for a noise-aware maximization scheme with a likelihood-based stopping criterion that promotes the proper noise-adapted fit given knowledge of the measurement noise $\sigma_{\mathbf{y}}$. Finally, we demonstrate comparable or improved performance against a wide selection of contemporary inverse solvers across multiple datasets, tasks, and noise levels.

## 1 INTRODUCTION

In this work, we study a broad class of problems involving the recovery of a signal $\mathbf{x}$ from a measurement

$$\mathbf{y} = \mathcal{A}(\mathbf{x}) + \boldsymbol{\eta}. \tag{1}$$

with noise $\boldsymbol{\eta}$ and measurement operator $\mathcal{A}$. Known as inverse problems, such formulations appear in a multitude of fields, with applications including acoustic reconstruction (Kac, 1966), seismic profiling (Hardage, 1985), X-ray computed tomography and magnetic resonance imaging (Suetens, 2017), and a large number of computer vision reconstruction tasks such as inpainting, deconvolution, colorization, super-resolution, and phase retrieval (Andrews and Hunt, 1977).

In many cases, $\mathcal{A}$ is assumed to be non-invertible[1], meaning that any solution $\mathbf{x}$ satisfying $\mathcal{A}(\mathbf{x}) = \mathbf{y}$ is not unique (Vogel, 2002). Moreover, due to noise in the measurement, it is often mathematically possible, but not practically desirable to fit perfectly to $\mathbf{y}$ for risk of *overfitting* to $\boldsymbol{\eta}$ (Aster et al., 2018). Therefore, a fundamental quandary in solving inverse problems is how one should select the best solution from an equivalence class of solutions, i.e., choosing $\mathbf{x}_* \in \{\mathbf{x} : \mathcal{A}(\mathbf{x}) \approx \mathbf{y}\}$.

In classical solvers, this is carried out by a regularizer on a normed error loss (Engl et al., 1996). One seeks

$$\mathbf{x}_* = \arg\min_{\mathbf{x}} R(\mathbf{x}) \quad \text{s.t.} \quad ||\mathcal{A}(\mathbf{x}) - \mathbf{y}|| \le \epsilon, \tag{2}$$

where $\epsilon$ is a soft error margin and $R$ is a simple function that satisfies user-specified heuristics, e.g., smoothness or total variation (Beck and Teboulle, 2009). However, such approaches often fail to produce realistic results, as $R$ lacks the ability to reconstruct details lost by $\mathcal{A}$. With the advent of deep

---

[1] We note there do exist a number of operators $\mathcal{A}$ of interest that are theoretically invertible, but practically non-invertible. For example, Gaussian blurs only become low-rank when the convolved image is truncated at the edges due to bounded image sizes.

generative models, practitioners found that restricting solutions to the range of a generative model $\mathbf{G}$ can greatly improve realism. Here, one may let $\mathbf{x} = G(\mathbf{w})$ and optimize over $\mathbf{w}$, which can be latent inputs (Bora et al., 2017) or weights (Ulyanov et al., 2018) of a deep neural network. Overall, these methods improve the fidelity of $\mathbf{x}$, but they lack interpretability and require a judiciously selected $R$ and $\epsilon$.

Recently, great strides have been made in solving inverse problems with diffusion models (Ho et al., 2020), which produce diverse, realistic samples (Dhariwal and Nichol, 2021; Esser et al., 2024) with robust generalization guarantees (Kadkhodaie et al., 2023). Moreover, they are interpretable, directly modeling the (Stein) score $\nabla \log p_t(\mathbf{x}_t)$. Sampling proceeds by reversing a noising process on $\mathbf{x}_0 \sim p_{\text{data}}$ roughly described (in black) by

$$\mathbf{x}_{t-1} = \texttt{denoise}[\mathbf{x}_t, \nabla \log p_\theta(\mathbf{x}_t)] + \texttt{guidance}. \tag{3}$$

Solvers then add a `guidance` term to lead $\mathbf{x}_t$ towards desirable solutions. While already effective, this approach suffers from a unique problem where a tractable form of the consistency error $||\mathcal{A}(\mathbf{x}) - \mathbf{y}||$ only exists for $\mathbf{x} = \mathbf{x}_0$ (Chung et al., 2022a). Such methods thus rely (explicitly or implicitly via (Song et al., 2020a)) on Tweedie's formula, which predicts $E(x_0|x_t)$, to estimate $\mathbf{x}_0$ given a noise prediction $\epsilon_{\texttt{corrected}}$

$$\hat{\mathbf{x}}_0[\epsilon_{\texttt{corrected}}] = \frac{1}{\sqrt{\alpha_t}} \left( \mathbf{x}_t - \sigma_t \epsilon_{\texttt{corrected}} \right). \tag{4}$$

This enables an estimate of $||\mathcal{A}(\mathbf{x}) - \mathbf{y}||$ at time $t$, which can produce a gradient that propagates the error back to $\mathbf{x}_t$.

We identify two issues with this framework in our work. First, we discover that the guidance obtained by this simple scheme can produce highly overfit models that generalize poorly given noisy measurements (Figures 4 and 5). Second, examining the conditions required for Eq. 4 to hold, we find that they are not generally true when using the *unconditional score* function $\nabla \log p_t(\mathbf{x}_t) \approx s_\theta(\mathbf{x}_t, t)$ and related quantity *unconditional noise* function $\epsilon_\theta(\mathbf{x}_t, t)$, modeled in general diffusion models (Section 3)[2]. On the other hand, we observe that it does hold when modeling the *data-conditional score* $\nabla \log p_t(\mathbf{x}_t|\mathbf{x}_0)$ and its related $\epsilon_{\texttt{corrected}}$. This term plays a crucial role during diffusion model training as a function of the data-dependent diffusion process centered at each $\mathbf{x}_0 \sim p_{\text{data}}$, but is generally intractable during sampling. Surprisingly, in inverse problems, the extra information present in $\mathbf{y}$ allows this term to be recovered to great accuracy by simple maximum likelihood estimation with the measurement model, thus allowing the estimation of a $\mathbf{y}$-conditional Tweedie's posterior $E(\mathbf{x}_0|\mathbf{x}_t, \mathbf{y})$.

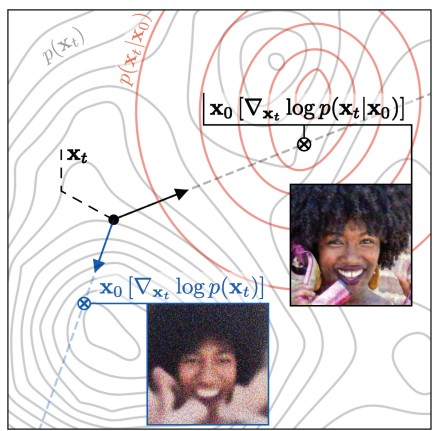

Figure 1: An **uncorrected** $\mathbf{x}_0$ **estimate** at time $t$ versus **our estimate**. Diffusion-based inverse problem solvers use an approximation of $\mathbf{x}_0$ to guide the diffusion process at each step (Section 3). However, using Tweedie's formula (Equation 4) with the score of the *unconditional* density $p(\mathbf{x}_t)$ may yield a low quality approximation of $\mathbf{x}_0$. To remedy this, we use the score of *data-conditional* density $p(\mathbf{x}_t|\mathbf{x}_0)$ obtained via a noise-aware maximum likelihood estimation framework (Section 4), yielding a superior estimate of $\mathbf{x}_0$.

**Contributions** We propose a novel frequentist's framework for solving inverse problems by directly sampling with a data-conditional score. We demonstrate that the maximum likelihood estimator for this score captures all the information present in the measurement $\mathbf{y}$, and propose a noise-aware maximization scheme to recover it even under significant measurement noise where many other algorithms fail (Figure 5). This data-conditional score can then be directly used during sampling in lieu of the unconditional score, resulting in a simple algorithm that requires no backpropagations through the neural function and is stable across noise levels and time steps, due to the noise-aware maximizer and linearity of the data-conditional diffusion respectively. Finally, we demonstrate

---

[2]Note that score functions $\nabla \log p_t(\mathbf{x}_t) \approx s_\theta(\mathbf{x}_t, t)$ and noise predictions $\epsilon_\theta(\mathbf{x}_t, t)$ are interchangeable via the relation $\epsilon_\theta = -\sigma_t s_\theta$.

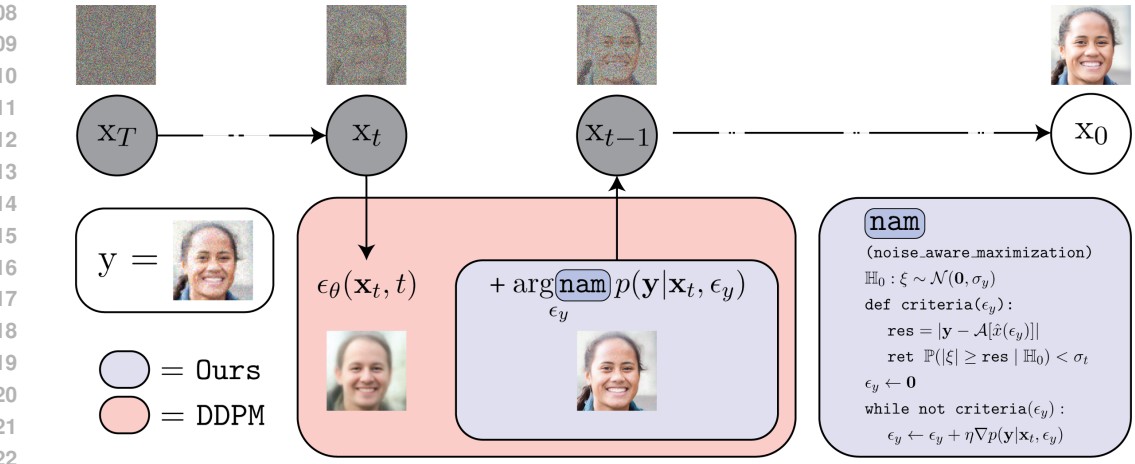

Figure 2: A demonstration of our proposed sampling algorithm on the super-resolution task. An initial noise prediction $\epsilon_\theta$ is corrected by the solution $\epsilon_{\mathbf{y}}$ of a **n**oise-**a**ware **m**aximization scheme of the measurement likelihood $p(\mathbf{y}|\mathbf{x}_t, \epsilon_y)$. This results in the corrected data-conditional noise prediction $(\epsilon_\theta + \epsilon_y) \approx -\sigma_t^{-1}\nabla \log p_t(\mathbf{x}_t|\mathbf{x}_0)$. For details see Section 4.

significant speed-ups over existing inverse solvers, while achieving state-of-the-art performance on a large selection of inverse problems, datasets and noise levels[3].

## 2 BACKGROUND AND RELATED WORK

### 2.1 DIFFUSION MODELS

Inspired by non-equilibrium thermodynamics, denoising diffusion probabilistic models (Ho et al., 2020) convert data $\mathbf{x}_0 \sim p_{\text{data}}(\mathbf{x})$ to noise $\mathbf{x}_T \sim \mathcal{N}(\mathbf{0}, \mathbf{I})$ via a diffusion process described by the variance-preserving stochastic differential equation (VP-SDE)

$$d\mathbf{x} = -\frac{\beta(t)}{2}\mathbf{x}dt + \sqrt{\beta(t)}d\mathbf{w}, \tag{5}$$

where $\beta(t) : \mathbb{R} \rightarrow [0,1]$ is a monotonically increasing noise schedule and $\mathbf{w}$ is the standard Wiener process (Song et al., 2020b). This leads to the marginal distribution

$$p_t(\mathbf{x}_t) = \mathbb{E}_{\mathbf{x}_0 \sim p_{\text{data}}}\big[\mathcal{N}(\mathbf{x}_t; \sqrt{\alpha_t}\mathbf{x}_0, \underbrace{(1-\alpha_t)}_{\sigma_t^2}\mathbf{I})\big], \quad \alpha_t = e^{-\frac{1}{2}\int_0^t \beta(s)ds}, \tag{6}$$

where $\mathcal{N}(\,\cdot\,;\mu,\Sigma)$ is the probability density function (pdf) of a normal distribution centered at $\mu$ with covariance $\Sigma$. Sampling from $p_{\text{data}}(\mathbf{x})$ can then occur by modeling the reverse diffusion, which has a simple form given by (Anderson, 1982)

$$d\overline{\mathbf{x}} = \left[-\frac{\beta(t)}{2}\mathbf{x} - \beta(t)\nabla_{\mathbf{x}}\log p_t(\mathbf{x}_t)\right]dt + \sqrt{\beta(t)}d\overline{\mathbf{w}}, \tag{7}$$

with reverse-time Wiener process $\overline{\mathbf{w}}$ and score function $\nabla_{\mathbf{x}}\log p_t(\mathbf{x}_t)$. Therefore, diffusion model training consists of approximating the score function with a model

$$s_\theta(\mathbf{x}_t, t) \approx \nabla_{\mathbf{x}}\log p_t(\mathbf{x}_t), \tag{8}$$

and sampling consists of obtaining solutions to the reverse-time SDE (7) with numerical solvers. A simple approach is given by the DDIM sampler with $\sigma_t = \sqrt{1 - \alpha_t}$ (Song et al., 2020a)

$$\mathbf{x}_{t-1} = \sqrt{\alpha_{t-1}}\frac{\mathbf{x}_t + \sigma_t^2\nabla \log p_t(\mathbf{x}_t)}{\sqrt{\alpha_t}} + \sigma_{t-1}\boldsymbol{\epsilon}. \tag{9}$$

---

[3]Code for method and experiments provided in `https://anonymous.4open.science/r/diffusion_conditional_sampling`

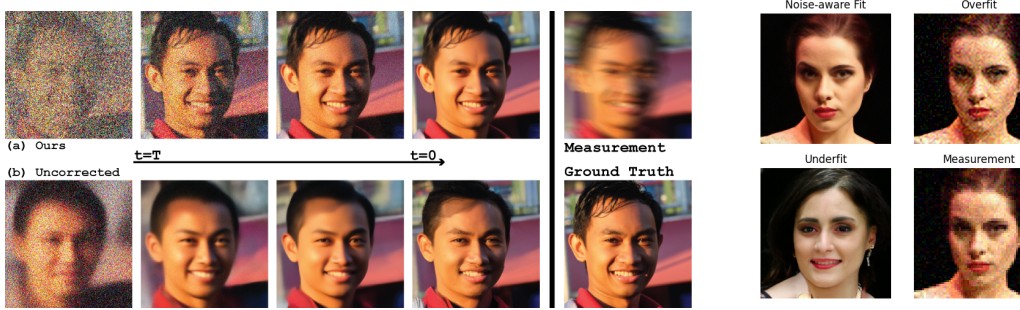

Figure 3: Estimated $\mathbf{x}_0$ given $\mathbf{x}_t$ at different times $t$ using a Tweedie's prediction of $\mathbf{x}_0$ with (a) the unconditional score versus (b) the data-conditional score via our maximum likelihood estimator. With the *unconditional* score, Tweedie's formula predicts the posterior mean of the dataset, rather than a sample $\mathbf{x}$ that satisfies $\mathcal{A}(\mathbf{x}) = \mathbf{y}$, especially at $T \gg 0$ (Section 3).

Figure 4: Comparison of goodness of fit for a super-resolution task. The best fit must balance between the measurement $\mathbf{y}$ and the data bias to achieve a good fit.

## 2.2 SOLVING INVERSE PROBLEMS WITH DIFFUSION MODELS

When solving inverse problems with diffusion models, the aim is to leverage information from $\mathbf{y}$ to define a **modified** reverse diffusion process

$$\mathbf{x}_T, \mathbf{x}_{T-1}, \ldots, \mathbf{x}_1, \mathbf{x}_0, \tag{10}$$

such that $\mathbf{x}_t$ coincides with the desired $\mathbf{x}$ (Eq. 1) precisely at $t = 0$. Previous approaches can generally be sorted into two categories, which we designate **posterior solvers** and **projection solvers**.

**Posterior Solvers** An intuitive approach is leveraging Bayes' rule to sample from the **posterior** distribution given a prior $p_t(\mathbf{x}_t)$ and observation $\mathbf{y}$:

$$\mathbf{x}_t \sim p(\mathbf{x}_t|\mathbf{y}) = \frac{p(\mathbf{y}|\mathbf{x}_t)p(\mathbf{x}_t)}{p(\mathbf{y})}. \tag{11}$$

Taking logs and gradients of both sides of the equation, we obtain a form of the conditional density that can be accurately approximated with the modeled score function

$$\nabla \log p(\mathbf{x}_t|\mathbf{y}) = \nabla \log p(\mathbf{y}|\mathbf{x}_t) + \nabla \log p(\mathbf{x}_t) \approx \nabla \log p(\mathbf{y}|\mathbf{x}_t) + s_\theta(\mathbf{x}_t, t), \tag{12}$$

and describes the core method of the DPS algorithm (Chung et al., 2022a). This strategy can also be extended to latent diffusion models, resulting in Latent-DPS and PSLD (Rout et al., 2023). Generally, the conditional term $\nabla \log p(\mathbf{y}|\mathbf{x}_t)$ cannot be exact due to reasons we will investigate subsequently in Section 3, though these approximations are improved in LGD (Song et al., 2023) and STSL (Rout et al., 2024). More recent works (Sun et al., 2024) propose an annealed Monte-Carlo-based perspective to posterior sampling, which results in a very similar algorithm to DPS. Much like MCG and ReSample (discussed in the next category), posterior solvers require estimating $\frac{\partial}{\partial \mathbf{x}_t}\mathbf{x}_0$ which involves backpropagation through the diffusion model, and significantly increases runtime and hampers scalability compared to unconditional sampling.

**Projection Solvers** Another approach involves guiding the reverse diffusion process by directly **projecting** $\mathbf{x}_t$ onto a manifold $\mathcal{M} = \{\mathbf{x} : \mathcal{A}(\mathbf{x}) = y\} \subseteq \mathbf{R}^d$ at each time step, i.e.

$$\mathbf{x}_t' = \mathbf{P}\hat{\mathbf{x}}_0[\mathbf{x}_t] \tag{13}$$

$$\mathbf{x}_{t-1} = \sqrt{\alpha_{t-1}}\frac{\mathbf{x}_t' + \sigma_t^2 \nabla \log p(\mathbf{x}_t'|\hat{\mathbf{x}}_0[\mathbf{x}_t])}{\sqrt{\alpha_t}} + \sigma_{t-1}\boldsymbol{\epsilon}. \tag{14}$$

Where $\hat{\mathbf{x}}_0[\mathbf{x}_t]$ is some prediction of $\mathbf{x}_0$ given only $\mathbf{x}_t$ (we elaborate in Section 3), and $\mathbf{P}$ is either a projection onto the low rank subspace or range of $\mathcal{A}$. The resulting algorithms are DDRM (Kawar et al., 2022) and DDNM (Wang et al., 2022), respectively. Of course, this strategy is often restricted

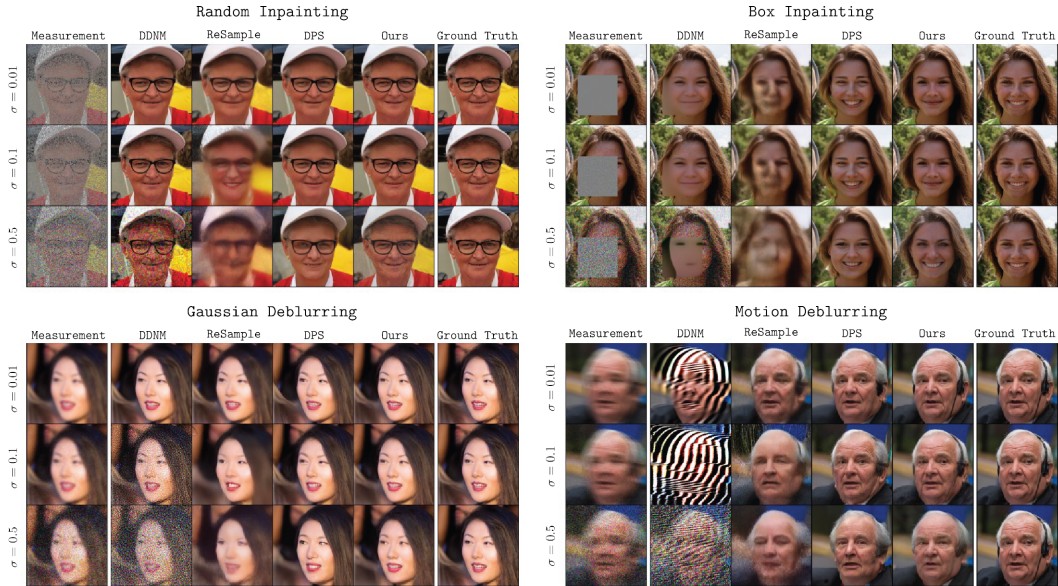

Figure 5: A demonstration of the variability in reconstruction quality across noise levels ($\sigma_{\mathbf{y}} \in \{0.01, 0.1, 0.5\}$) of many diffusion-based solvers. While DPS approaches the robustness of our method, it is significantly more expensive, requiring gradients of the score network and more than $3\times$ memory cost. More examples in Appendix D.

to situations where two conditions simultaneously hold true: (1) the measurement operator $\mathcal{A}$ is linear, and (2) the inverse problem is noiseless, i.e, $\eta$ is identically $\mathbf{0}$. These assumptions drastically limit the applicability of such models. The linearity restriction can be lifted by taking derivatives the measurement discrepancy, as in MCG (Chung et al., 2022b) and ReSample (Song et al., 2024), though this comes at the cost of significantly increased computation, requiring $\frac{\partial}{\partial \mathbf{x}_t} \mathbf{x}_0$ which involves backpropagating through the score network. Finally, (Cardoso et al., 2023) straddles the line between both categories — while MCGdiff is ostensibly a Bayesian solver, it bears greater resemblance to projection solvers since it does not form the decomposition in Eq. 12 and also samples by projecting each iterate to the null-space of $\mathcal{A}$, thus implementing a projected n-particle sequential monte carlo (SMC) sampling algorithm.

**A Maximum Likelihood Solver**   We take a different perspective on solving the inverse problem. As seen in Section 3, both **projection** and **posterior** solvers must quantify the discrepancy between $\mathbf{x}_t$ and $\mathbf{y}$ via the consistency error $||\mathcal{A}(\mathbf{x}_0) - \mathbf{y}||$ at each diffusion step. Due to the complexity of the diffusion process, this involves approximating a fundamentally intractable quantity. In Section 4, we construct a simpler process whose parameters can be obtained via maximum likelihood estimation. Unlike the evidence lower bound proposed in (Mardani et al., 2023), we derive an explicit likelihood model, which is amenable to an optimization scheme with a probabilistic noise-aware stopping criterion. Finally, we show that the resulting algorithm is simple, fast, and adaptable to noise.

## 3   REVISITING TWEEDIE'S FOR INVERSE PROBLEMS

Diffusion-based inverse problem solvers (Section 2.2) face a fundamental *computability paradox*: since the consistency error is only explicitly known at $t = 0$ via the likelihood function

$$p(\mathbf{y}|\mathbf{x}_0) \propto \exp\left(-\frac{1}{2\sigma_{\mathbf{y}}^2}||\mathbf{y} - \mathcal{A}(\mathbf{x}_0)||_2^2\right) \tag{15}$$

we cannot exactly guide the diffusion process $d\mathbf{x}_t$ at time $t > 0$ without solving for $\mathbf{x}_0$. However, we also cannot generally obtain $\mathbf{x}_0$ without first computing $\mathbf{x}_t$. Accurately estimating $\hat{\mathbf{x}}_0 \approx \mathbf{x}_0$ is a fundamental problem all solvers must contend with to function properly.

| **Algorithm 1** Diffusion Conditional Sampler | **Algorithm 2** Noise-aware Maximization (`nam`) |
|---|---|
| 1: **Input:** $\mathbf{y}, \mathcal{A}, \epsilon_\theta$ \| **Output:** $\mathbf{x}_0$ | 1: **Input:** $\mathbf{y}, \mathcal{A}, \mathbf{x}_t, \epsilon_\theta$ \| **Output:** $\epsilon_y$ |
| 2: $\mathbf{x}_T \sim \mathcal{N}(\mathbf{0}, \mathbf{I})$ | 2: $\epsilon_y \leftarrow \mathbf{0}$ |
| 3: **for** $t = T$ to $1$ **do** | 3: $\hat{\mathbf{x}} \leftarrow \texttt{Tweedie's}(\mathbf{x}_t, \epsilon_\theta + \epsilon_y)$ |
| 4: $\quad \boldsymbol{\epsilon} \sim \mathcal{N}(\mathbf{0}, \mathbf{I})$ | 4: **while** $2\Phi[-\|y - \mathcal{A}[\hat{\mathbf{x}}]\|_1^1/(d\sigma_\mathbf{y})] < \sigma_t$ **do** |
| 5: $\quad \epsilon_\mathbf{y} \leftarrow \underset{\epsilon_\mathbf{y}}{\arg\texttt{nam}}\, p_t\left(\mathbf{y}\Big\| \frac{\mathbf{x}_t + \sigma_t^2(\epsilon_\theta + \epsilon_\mathbf{y})}{\sqrt{\alpha_t}}\right)$ | 5: $\quad \epsilon_y \leftarrow \epsilon_y + \eta\nabla\log p_t\left(\mathbf{y}\Big\|\frac{\mathbf{x}_t + \sigma_t^2(\epsilon_\theta + \epsilon_\mathbf{y})}{\sqrt{\alpha_t}}\right)$ |
| 6: $\quad \mathbf{x}_{t-1} \leftarrow \sqrt{\alpha_{t-1}}\frac{\mathbf{x}_t - \sigma_t(\epsilon_\theta + \epsilon_y)}{\sqrt{\alpha_t}} + \sigma_{t-1}\boldsymbol{\epsilon}$ | 6: $\quad \hat{\mathbf{x}} \leftarrow \texttt{Tweedie's}(\mathbf{x}_t, \epsilon_\theta + \epsilon_y)$ |
| 7: **end for** | 7: **end while** |

In **posterior** solvers, this culminates in the computation of $\nabla \log p(\mathbf{y}|\mathbf{x}_t)$, which is approximated by $\nabla \log p(\mathbf{y}|\hat{\mathbf{x}}_0)$. In **projection** solvers, this is the projection step $\mathbf{P}\mathbf{x}_t$, which is driven by a projection on $\hat{\mathbf{x}}_0$, followed by a DDIM step (Song et al., 2020a) that involves $\hat{\mathbf{x}}_0$. In both cases, one turns to an estimator based on Tweedie's formula, which provides a simple approximation for $\mathbf{x}_0$ given the current $\mathbf{x}_t$.

**Lemma 3.1** (An approximation of $\mathbf{x}_0$ inspired by Tweedie's formula). *Let $\mathbf{x}_0$ be given. Suppose $\mathbf{x}_t$ is distributed as*

$$p_t(\mathbf{x}_t|\mathbf{x}_0) = \mathcal{N}(\mathbf{x}_t; \sqrt{\alpha_t}\mathbf{x}_0, \underbrace{1 - \alpha_t}_{\sigma_t^2}\mathbf{I}). \tag{16}$$

*Then $\mathbf{x}_0$ can be recovered via*

$$\mathbf{x}_0 = \frac{1}{\sqrt{\alpha_t}}\left[\mathbf{x}_t + \sigma_t^2 \nabla_{\mathbf{x}_t} \log p_t(\mathbf{x}_t|\mathbf{x}_0)\right]. \tag{17}$$

A key detail in the above statement is that predicting $\mathbf{x}_0$ with Tweedie's formula requires a normally distributed $\mathbf{x}_t$, rather than the (usually highly multi-modal) data distribution $p_{\text{data}}$ modeled by a diffusion model (or its noisy counterpart, convolved against a normal distribution with variance $\sigma_t^2$). The reliance on this assumption becomes clear in the simple proof (in Appendix A) — observe that the cancellations in the last equality *require* the linear form of the Gaussian score to hold true. In fact, this is a necessary and sufficient condition for Eq. 17 to hold.

**Theorem 3.2.** *Tweedie's formula predicts $\mathbf{x}_0$ **if and only if** $\mathbf{x}_t$ is distributed as a simple isotropic Gaussian.*

In practice, Theorem 3.2 exposes potential sources of instability which may arise when Tweedie's formula is directly used without adjustment to approximate the endpoint of the reverse process. While at $t \approx 0$, $p_t(\mathbf{x}_0|\mathbf{x}_t)$ may approach an isotropic Gaussian, at large $t \gg 0$ we expect Tweedie's to instead predict the expectation over all "nearby" data, i.e., the posterior mean:

$$\mathbb{E}_{\mathbf{x}_t \sim p_t}[\mathbf{x}_0|\mathbf{x}_t], \tag{18}$$

with a neighborhood that grows to encompass all $p_{\text{data}}$ itself at $t \approx T$. This phenomenon is visible in Figure 3, where at larger values of $t$, the fidelity of the estimated $\mathbf{x}_0$ is very poor, indicating less stable sampling. The underlying reason is that this estimator of $\mathbf{x}_0$ cannot be a sufficient statistic for $(\mathbf{x}_0|\mathbf{x}_t, \mathbf{y})$, but rather of only $(\mathbf{x}_0|\mathbf{x}_t)$, since $\mathbf{y}$ is never considered in the approximation. Therefore, the information in $\mathbf{y}$ could still be leveraged for improving this approximation in the context of our inverse problem task — and carefully in the noisy regime to prevent overfitting. This motivates the method we outline in the following section.

## 4 DIFFUSION CONDITIONAL SAMPLING

We propose **Diffusion Conditional Sampling (DCS)**, a novel framework for solving inverse problems with diffusion models. We sample from the solution set $\{\mathbf{x} : \mathcal{A}[\mathbf{x}] = \mathbf{y}\}$ of an inverse problem by leveraging a noise-aware maximization scheme, and obtain the maximum likelihood estimator of a simple single-parameter noisy measurement model. This measurement model is formed by

combining Eqs. 15 and 17, resulting in a closed form expression in terms of the *data-conditional score* $\nabla \log p_t(\mathbf{x}_t|\mathbf{x}_0)$ and consistency error $||\mathcal{A}(\mathbf{x}) - \mathbf{y}||$ at each step:

$$\log p(\mathbf{y}|\mathbf{x}_0(\epsilon_{\mathbf{y}}, \mathbf{x}_t)) \propto -\frac{1}{2\sigma_{\mathbf{y}}^2} \left\| y - \mathcal{A}\left( \frac{1}{\sqrt{\alpha_t}}[\mathbf{x}_t + \sigma_t^2 \nabla_{\mathbf{x}_t} \log p_t(\mathbf{x}_t|\mathbf{x}_0)] \right) \right\|_2^2. \quad (19)$$

We note that $p_t(\mathbf{x}_t|\mathbf{x}_0)$ is a Gaussian distribution, meaning that the application of Tweedie's formula in Eq. 19 will exactly recover $\mathbf{x}_0$ (Theorem 3.2). Thus, defining

$$\nabla \log p_t(\mathbf{x}_t|\mathbf{x}_0) = -\sigma_t^{-1}[\epsilon_\theta(\mathbf{x}_t, t) + \epsilon_{\mathbf{y}}], \quad (20)$$

we can solve for our single parameter $\epsilon_y$ by maximizing the joint likelihood between the measurement $\mathbf{y}$ and our parameter $\epsilon_{\mathbf{y}}$. This forms our data-conditional score estimate $s_{\text{corrected}} =$

$$-\sigma_t^{-1} \left[ \epsilon_\theta(\mathbf{x}_t, t) + \arg\max_{\epsilon_{\mathbf{y}}} -\frac{1}{2\sigma_{\mathbf{y}}^2} \left\| y - \mathcal{A}\left( \frac{1}{\sqrt{\alpha_t}}(\mathbf{x}_t - \sigma_t[\epsilon_\theta(\mathbf{x}_t, t) + \epsilon_{\mathbf{y}}]) \right) \right\|_2^2 \right], \quad (21)$$

of the true data-conditional score $\nabla \log p_t(\mathbf{x}_t|\mathbf{x}_0)$. This can be interchanged with the corrected noise prediction via the relation $\epsilon_{\text{corrected}} = -\sigma_t s_{\text{corrected}}$.

Given that Eq. 21 is an ill-posed optimization problem, we seek to sample from the solution set $\{\mathbf{x} : \mathcal{A}[\mathbf{x}] = \mathbf{y}\}$ given by the measurement $\mathbf{y}$ through a noise-aware maximization algorithm, which we outline below. We then leverage our learned parametric model to sample $\mathbf{x}_{t-1}$ via the standard DDPM sampling algorithm (Ho et al., 2020). Applying this step to each $t = T, \ldots, 1$, we arrive at our proposed DCS algorithm. Our approach is summarized in Algorithm 1. We note that it is remarkably simple, and easily modified from the unconditional sampler in DDPM (Ho et al., 2020). Additional details can be found in Appendix C. Below, we discuss two critical components of our proposed algorithm.

**Noise-aware Maximization**   We propose a **n**oise-**a**ware **m**aximization scheme (nam) to improve stability across noise levels. As previously discussed, we seek the data-conditional score (Eq. 21), which can be understood as the maximum likelihood solution to the measurement model (Eq. 19).

However, given a single noisy measurement $\mathbf{y} = \mathcal{A}[\mathbf{x}] + \eta$, there is a high risk of overfitting to noise $\eta$ (Figures 4 and 5). To mitigate this problem, we propose a maximization scheme with a specialized early stopping criterion based on the measurement likelihood. We leverage the intuition that the corrected data-conditional score should yield a prediction via Eq. 17 whose residual

$$\texttt{res} = \mathbf{y} - \mathcal{A}[\hat{\mathbf{x}}_0] \quad (22)$$

is normally distributed with variance $\sigma_{\mathbf{y}}^2$. In other words, $\texttt{res}$ should come from the same distribution as $\eta$. Let this be the *null hypothesis* $\mathbb{H}_0$ — we thus seek to end the likelihood maximization process as soon as $\mathbb{H}_0$ holds. Specifically, we optimize Eq. 19 until the likelihood of the *alternate hypothesis* $\mathbb{H}_1$, that $\texttt{res}$ is *not* distributed as $\eta$, is below a desired threshold $p_{\text{critical}}$. Since overfitting is more problematic at the end of sampling ($t \approx 0$) than the beginning of sampling ($t \approx T$), we set $p_{\text{critical}}$ dynamically as a function of $t$, namely $p_{\text{critical}}(t) = \sigma_t$. This scheme is heavily inspired by the classical two-sided **z**-test (Hogg et al., 2013) with $d$ samples, where $d$ is the dimensionality of the image. Formally, we use the early-stopping criterion at each time $t$

$$P(|\xi| > |\texttt{res}| \,|\, \mathbb{H}_0) = 2\Phi(-|\texttt{res}|/\sigma_{\mathbf{y}}) < \sigma_t, \quad (23)$$

where $\xi_i \overset{\text{iid}}{\sim} \mathcal{N}(\mathbf{0}, \sigma_{\mathbf{y}}^2)$ and $\Phi$ is the CDF of a standard normal distribution. The full noise-aware maximization algorithm can be summarized by Algorithm 2. Since our loss function (Eq. 19) is quadratic, our proposed nam has worst-case linear convergence guarantees due to classical results in gradient descent (Boyd and Vandenberghe, 2004; Ryu and Boyd, 2016).

**Sufficiency**   We rigorously investigate the conditions under which **DCS** captures all the signal present in the measurement $\mathbf{y}$. Formally, we find that $s_{\text{corrected}}$ (resp. $\epsilon_{\text{corrected}}$) is statistically sufficient for $\mathbf{y}$. Letting $\mathbf{f}(\mathbf{y})$ be the function that obtains $\nabla \log p_t(\mathbf{x}_t|\mathbf{x}_0)$ via Eq. 21, we show that $\mathbf{y}$ is measurable under the $\sigma$-algebra induced by the measurement $\mathbf{f}$. Intuitively, we demonstrate that $\mathbf{f}(\mathbf{y})$ contains as much information as possible about the underlying signal $\mathbf{x}_0$ as can be gathered via $\mathbf{y}$. The theoretical and intuitive statements can be summarized by the simple conditional equivalence

$$p(\mathbf{y}|\epsilon_\theta + \epsilon_{\mathbf{y}}) = p(\mathbf{y}|\mathbf{x}_0). \quad (24)$$

We prove in Theorem $A.3$, that $\epsilon_{\mathbf{y}}$ is a sufficient statistic for $\mathbf{x}_0$ with measurement $\mathbf{y}$ under mild regularity conditions on $\mathcal{A}$ and $\boldsymbol{\eta}$.

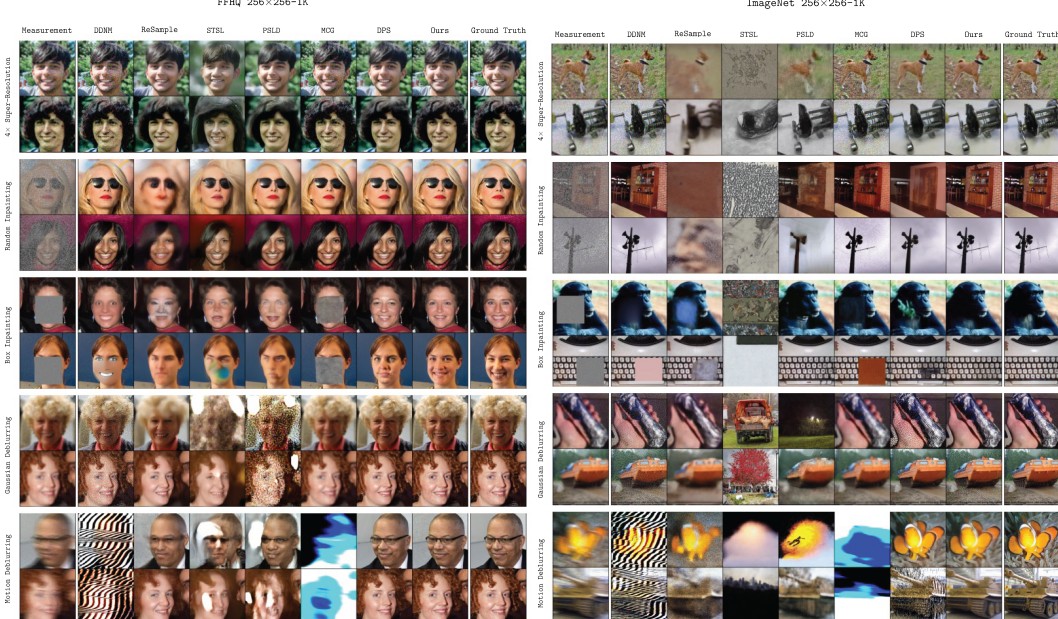

Figure 6: Qualitative comparison of our proposed method against competing works on FFHQ 256×256-1K (left) and ImageNet 256×256-1K (right). Further comparisons can be found in Appendix D.

## 4.1 EFFICIENCY

We discuss the computational efficiency of our algorithm in two respects: removing the need to compute expensive gradients of the score function, and improved convergence due to the linearity of the data-conditional diffusion process.

**No Expensive $\nabla s_\theta(\mathbf{x}_t, t)$ Evaluations** A drawback of many existing algorithms is the need to compute gradients of the score network during sampling (Table 2). This is the most expensive computation in the diffusion step, increasing the runtime of the algorithm by **2-3×**. However, this is unavoidable in posterior solvers. Projection solvers sidestep this issue by framing a diffusion process in a subspace of $\mathcal{A}$ — however, this cannot be done when $\mathcal{A}$ is nonlinear. To our knowledge, our algorithm is the only algorithm that can handle nonlinear operators without requiring backpropagations through the score network. We note that the most similar algorithm is ReSample. However, as discussed in Appendix C.4, ReSample still requires backpropagations in its implementation, even though this is not discussed in the paper.

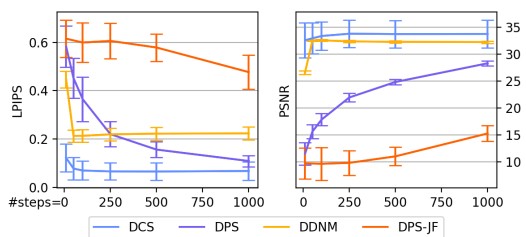

Figure 7: A study on the effect of $T$ on solver performance. While other approaches exhibit poor performance due to the nonlinearity of the original reverse diffusion process, our method remains nearly invariant to $T$ due to the near-linearity of the *data-conditional* diffusion process.

**A Near-Linear Reverse Process** As **DCS** models $\nabla \log p_t(\mathbf{x}_t|\mathbf{x}_0)$, it is able to sample approximately from the *data-conditional* reverse diffusion process, which reverses the forward process defined in Eq. 6. In the ideal scenario, this process is Gaussian, meaning that Tweedie's exactly recovers $\mathbf{x}_0$, and the diffusion process can be solved in a single step. In reality, our approximation of this process is correct up to the information about $\mathbf{x}_0$ present in $\mathbf{y}$ (Theorem A.3), under the assumptions detailed in the previous section.

In Figure 7, we experimentally validate the robustness of our algorithm to the total diffusion steps ($T$) with the super-resolution task on a subset of the FFHQ $256 \times 256$ dataset. We compare against DPS, DPS-JF, and DDNM at $\sigma_{\mathbf{y}} = 0.05$.

| FFHQ | SR ×4 | | | Random Inpainting | | | Box Inpainting | | | Gaussian Deblurring | | | Motion Deblurring | | |
|---|---|---|---|---|---|---|---|---|---|---|---|---|---|---|---|
| $\sigma_\mathbf{y} = 0.01$ | LPIPS↓ | PSNR↑ | FID↓ | LPIPS↓ | PSNR↑ | FID↓ | LPIPS↓ | PSNR↑ | FID↓ | LPIPS↓ | PSNR↑ | FID↓ | LPIPS↓ | PSNR↑ | FID↓ |
| Ours | **0.137** | **30.138** | **19.45** | **0.024** | **34.839** | 21.19 | **0.088** | **25.112** | **19.25** | **0.103** | **28.688** | 22.62 | **0.087** | **29.480** | **26.67** |
| DPS | 0.163 | 25.908 | 33.21 | 0.105 | 29.539 | 29.72 | 0.113 | 23.521 | 24.41 | 0.129 | 26.484 | 26.85 | 0.159 | 24.411 | 29.84 |
| DPS-JF | 0.488 | 14.193 | 44.98 | 0.335 | 19.566 | 58.45 | 0.178 | 20.118 | 28.10 | 0.211 | 23.063 | 34.42 | 0.289 | 19.927 | 40.94 |
| DPS-JF ($T = 100$) | 0.589 | 9.473 | 41.24 | 0.578 | 10.072 | 42.06 | 0.571 | 10.618 | 43.08 | 0.563 | 10.859 | 43.77 | 0.566 | 10.922 | 41.26 |
| LGD-MC-JF | 0.566 | 10.502 | 41.25 | 0.537 | 12.154 | 43.85 | 0.497 | 13.811 | 46.40 | 0.452 | 15.569 | 46.22 | 0.457 | 15.446 | 46.08 |
| LGD-MC-JF ($T = 100$) | 0.593 | 9.346 | 40.60 | 0.587 | 9.688 | 40.99 | 0.581 | 10.126 | 42.30 | 0.574 | 10.273 | 40.59 | 0.574 | 10.364 | 40.51 |
| MCG | 0.144 | 24.838 | 31.47 | 0.073 | 30.592 | 22.22 | 0.453 | 15.444 | 185.54 | 0.209 | 23.512 | 67.88 | 0.217 | 22.930 | 292.13 |
| DDNM | 0.208 | 26.277 | 51.33 | 0.040 | 33.076 | 23.35 | 0.209 | 18.118 | 88.32 | 0.235 | 26.086 | 71.47 | 0.424 | 14.221 | 250.92 |
| DDRM | 0.502 | 13.002 | 222.45 | 0.393 | 15.935 | 163.91 | 0.472 | 12.148 | 209.18 | - | - | - | - | - | - |
| Latent-DPS | 0.324 | 20.086 | 100.27 | 0.249 | 22.64 | 297.43 | 0.227 | 22.184 | 211.23 | 0.390 | 25.608 | 321.5 | 0.950 | -6.753 | 354.95 |
| PSLD | 0.311 | 20.547 | 42.26 | 0.250 | 22.84 | 214.08 | 0.221 | 22.23 | 204.87 | 0.200 | 23.77 | 318.20 | 0.213 | 23.277 | 359.40 |
| STSL | 0.614 | 16.063 | 327.38 | 0.476 | 17.859 | 190.64 | 0.436 | 11.843 | 190.64 | 0.583 | 15.196 | 364.07 | 0.604 | 10.095 | 388.68 |
| ReSample | 0.221 | 24.699 | 48.87 | 0.467 | 22.488 | 96.89 | 0.247 | 20.852 | 50.3 | 0.191 | 27.151 | 46.5 | 0.281 | 25.138 | 65.06 |

| FFHQ | SR ×4 | | | Random Inpainting | | | Box Inpainting | | | Gaussian Deblurring | | | Motion Deblurring | | |
|---|---|---|---|---|---|---|---|---|---|---|---|---|---|---|---|
| $\sigma_\mathbf{y} = 0.1$ | LPIPS↓ | PSNR↑ | FID↓ | LPIPS↓ | PSNR↑ | FID↓ | LPIPS↓ | PSNR↑ | FID↓ | LPIPS↓ | PSNR↑ | FID↓ | LPIPS↓ | PSNR↑ | FID↓ |
| Ours | 0.1748 | 24.879 | 30.107 | 0.1490 | 27.536 | 32.800 | 0.1631 | 23.217 | 26.444 | 0.1763 | 25.955 | 26.083 | 0.2238 | 24.612 | 31.400 |
| DPS | 0.1847 | 24.786 | 35.455 | 0.1566 | 26.717 | 35.238 | 0.1583 | 22.576 | 32.469 | 0.1797 | 24.720 | 33.530 | 0.2107 | 22.412 | 35.086 |
| DPS-JF | 0.494 | 14.111 | 46.59 | 0.371 | 18.310 | 56.49 | 0.226 | 19.451 | 34.02 | 0.246 | 21.808 | 35.53 | 0.342 | 18.339 | 40.70 |
| DPS-JF ($T = 100$) | 0.589 | 9.432 | 40.82 | 0.582 | 9.900 | 39.58 | 0.572 | 10.552 | 42.90 | 0.564 | 10.894 | 42.36 | 0.568 | 10.943 | 42.44 |
| LGD-MC-JF | 0.557 | 11.208 | 44.86 | 0.511 | 13.265 | 49.07 | 0.452 | 15.243 | 48.68 | 0.396 | 17.434 | 46.76 | 0.400 | 17.301 | 45.53 |
| LGD-MC-JF ($T = 100$) | 0.594 | 9.324 | 41.06 | 0.589 | 9.655 | 41.65 | 0.580 | 10.107 | 42.97 | 0.578 | 10.334 | 41.84 | 0.574 | 10.312 | 41.53 |
| MCG | 0.5464 | 20.441 | 102.60 | 0.2272 | 26.000 | 50.403 | 0.5791 | 15.297 | 207.23 | 0.4293 | 25.801 | 69.287 | 0.9729 | -7.104 | 295.32 |
| DDNM | 0.6230 | 21.493 | 145.889 | 0.179 | 24.964 | 39.183 | 0.334 | 19.195 | 72.105 | 1.220 | 10.727 | 176.756 | 0.739 | 5.099 | 524.021 |
| DDRM | 0.7853 | 6.3273 | 271.70 | 0.6018 | 10.995 | 255.95 | 0.6323 | 9.6360 | 288.11 | - | - | - | - | - | - |
| Latent-DPS | 0.3444 | 19.971 | 45.052 | 0.4455 | 18.117 | 109.83 | 0.6410 | 11.365 | 326.75 | 0.6398 | 13.762 | 330.93 | 0.6360 | 12.524 | 334.43 |
| PSLD | 0.3481 | 19.251 | 47.864 | 0.3105 | 20.588 | 41.737 | 0.3121 | 19.874 | 40.428 | 0.2897 | 21.068 | 36.600 | 0.3307 | 19.224 | 40.374 |
| STSL | 0.3161 | 20.279 | 40.163 | 0.3722 | 19.247 | 54.648 | 0.5481 | 13.864 | 183.00 | 0.5137 | 16.411 | 169.32 | 0.5188 | 15.463 | 163.65 |
| ReSample | 0.2613 | 24.184 | 50.224 | 0.5267 | 21.575 | 103.62 | 0.2789 | 20.581 | 53.263 | 0.2984 | 23.980 | 56.489 | 0.6456 | 19.912 | 110.42 |

| ImageNet | SR ×4 | | | Random Inpainting | | | Box Inpainting | | | Gaussian Deblurring | | | Motion Deblurring | | |
|---|---|---|---|---|---|---|---|---|---|---|---|---|---|---|---|
| $\sigma_\mathbf{y} = 0.01$ | LPIPS↓ | PSNR↑ | FID↓ | LPIPS↓ | PSNR↑ | FID↓ | LPIPS↓ | PSNR↑ | FID↓ | LPIPS↓ | PSNR↑ | FID↓ | LPIPS↓ | PSNR↑ | FID↓ |
| Ours | **0.238** | 23.452 | 39.41 | 0.142 | 26.063 | 34.46 | **0.230** | 20.625 | 37.11 | **0.253** | 24.218 | 38.96 | **0.203** | 24.619 | 38.63 |
| DPS | 0.309 | 23.994 | 49.81 | 0.266 | 25.054 | 38.87 | 0.301 | 18.764 | 34.85 | 0.493 | 19.138 | 61.59 | 0.460 | 18.645 | 53.21 |
| MCG | 0.638 | 15.619 | 89.39 | 0.198 | 24.343 | 35.19 | 0.273 | 16.675 | 80.35 | 0.643 | 21.177 | 124.61 | 0.980 | -5.726 | 231.11 |
| DDNM | 0.333 | 25.159 | 51.33 | 0.084 | 28.345 | 20.27 | 0.258 | 17.424 | 85.41 | 0.456 | 24.351 | 67.98 | 0.694 | 5.721 | 304.21 |
| DDRM | 0.907 | 6.592 | 277.81 | 0.835 | 10.145 | 215.77 | 0.758 | 11.695 | 198.83 | - | - | - | - | - | - |
| Latent-DPS | 0.642 | 17.973 | 144.82 | 0.603 | 19.881 | 144.81 | 0.751 | 11.964 | 138.33 | 0.805 | 10.532 | 139.62 | 0.821 | 10.697 | 150.49 |
| PSLD | 0.380 | 22.690 | 168.08 | 0.306 | 24.167 | 125.25 | 0.330 | 18.290 | 156.30 | 0.397 | 23.076 | 134.18 | 0.453 | 21.576 | 187.21 |
| STSL | 0.617 | 19.682 | 143.62 | 0.599 | 20.500 | 137.09 | 0.832 | 9.560 | 170.93 | 0.869 | 8.708 | 183.38 | 0.882 | 8.527 | 195.74 |
| ReSample | 0.552 | 20.260 | 133.42 | 0.820 | 17.775 | 229.82 | 0.504 | 16.795 | 138.97 | 0.513 | 21.578 | 116.04 | 0.573 | 20.430 | 145.67 |

| ImageNet | SR ×4 | | | Random Inpainting | | | Box Inpainting | | | Gaussian Deblurring | | | Motion Deblurring | | |
|---|---|---|---|---|---|---|---|---|---|---|---|---|---|---|---|
| $\sigma_\mathbf{y} = 0.1$ | LPIPS↓ | PSNR↑ | FID↓ | LPIPS↓ | PSNR↑ | FID↓ | LPIPS↓ | PSNR↑ | FID↓ | LPIPS↓ | PSNR↑ | FID↓ | LPIPS↓ | PSNR↑ | FID↓ |
| Ours | **0.4015** | 22.988 | 48.211 | **0.1655** | 26.043 | 34.469 | **0.2428** | 19.697 | 46.026 | **0.4068** | 22.283 | 51.131 | **0.4348** | 20.428 | 61.48 |
| DPS | 0.5397 | 18.630 | 85.063 | 0.5056 | 20.101 | 82.737 | 0.4789 | 18.033 | 83.059 | 0.4124 | 20.566 | 65.066 | 0.4499 | 18.905 | 75.652 |
| MCG | 0.8858 | 14.008 | 145.06 | 0.4591 | 19.915 | 78.863 | 0.4327 | 15.634 | 123.96 | 0.6502 | 22.004 | 117.43 | 0.9836 | -6.868 | 231.30 |
| DDNM | 0.7509 | 20.978 | 133.28 | 0.1693 | 25.634 | 35.718 | 0.4001 | 18.064 | 110.78 | 1.2209 | 9.6021 | 202.74 | 0.7825 | 5.0091 | 350.13 |
| DDRM | 0.9852 | 5.9810 | 425.77 | 0.9365 | 7.3908 | 358.10 | 0.8412 | 8.6456 | 240.95 | - | - | - | - | - | - |
| Latent-DPS | 0.7257 | 15.676 | 147.65 | 0.7973 | 9.4153 | 146.69 | 0.7980 | 9.3345 | 146.51 | 0.7988 | 9.3032 | 193.84 | 0.8525 | 9.1369 | 170.08 |
| PSLD | 0.4731 | 20.875 | 130.99 | 0.6068 | 19.668 | 145.51 | 0.7028 | 13.909 | 146.74 | 0.7372 | 14.181 | 139.90 | 0.7504 | 13.767 | 149.75 |
| ReSample | 0.6514 | 18.997 | 155.26 | 0.9654 | 13.612 | 281.82 | 0.5980 | 15.843 | 168.06 | 0.6814 | 19.233 | 173.72 | 1.0461 | 15.249 | 223.52 |

Table 1: Quantitative experiments on FFHQ 256x256-1K and ImageNet-1K datasets across various inverse problem tasks and noise levels ($\sigma_\mathbf{y} \in \{0.01, 0.1\}$). We compare against pixel-based solvers (upper half) and latent-based solvers (lower half).

## 5 EXPERIMENTS

We examine the empirical performance of **DCS** across a variety of natural image based inverse problems. We build a baseline by comparing across a range of state-of-the-art methods that operate in the pixel space and methods which employ latent diffusion models, all detailed in Table 2.

Quantitatively, we use a set of metrics to evaluate the quality of signal recovery: Learned Perceptual Image Patch Similarity (LPIPS), peak signal-to-noise ratio (PSNR), and Frechet Inception Distance (FID).

We run **DCS** and the other methods listed in Table 2 on the FFHQ-256 (Karras et al., 2019), (Kazemi and Sullivan, 2014), ImageNet (Deng et al., 2009), and CelebA-HQ (Liu et al., 2015) datasets. For FFHQ and CelebA-HQ, we use the pretrained FFHQ model weights from (Chung et al., 2022a) for pixel space models, and the pretrained FFHQ model with a VQ-F4 first stage model (Rombach et al., 2022) in latent space models. For ImageNet, we again use pretrained model weights from (Chung et al., 2022a) in pixel-based diffusion solvers, and the Stable Diffusion v1.5 latent model for latent solvers.

We examine five operator inversion tasks: Super-Resolution, Gaussian Deblurring, Motion Deblurring, Random Inpainting, and Box Inpainting. All experiments were run with additive Gaussian noise with standard deviation $\sigma_\mathbf{y} = 0.01$ (we present results at a higher noise level in Section 5.1). We also present quantitative results on FFHQ and ImageNet in Table 1, and a qualitative comparison in Figure 6. We delegate experiments on CelebA, subsets of FFHQ used in other works, further qualitative comparisons, and details of the implementation to Appendix B, C and D.

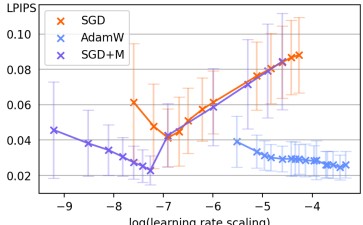

Figure 8: Comparison of **DCS** performance with different optimizers. LPIPS score of the predicted $\mathbf{x}_0$ images is plotted against the natural log of learning rate scaling factor for each optimizer.

| Solver | Type | Space | No NFE Backprop | Runtime | Memory |
|---|---|---|---|---|---|
| DPS (Chung et al., 2023) | Posterior | Pixel | ✗ | 6x | 3.2x |
| DPS-JF | Posterior | Pixel | ✓ | 1.5x | 1.1x |
| LGD-MC-JF (n=10) Song et al. (2023) | Posterior | Pixel | ✗ | 2x | 1.1x |
| MCG (Chung et al., 2022b) | Projection | Pixel | ✗ | 6.1x | 3.2x |
| DDNM (Wang et al., 2022) | Projection | Pixel | ✓ | 1.75x | 1x |
| DDRM (Kawar et al., 2022) | Projection | Pixel | ✓ | 1.75x | 1x |
| Latent-DPS[4] | Posterior | Latent | ✗ | 6.1x | 8.9x |
| PSLD (Rout et al., 2023) | Posterior | Latent | ✗ | 7.5x | 15x |
| STSL (Rout et al., 2024) | Posterior | Latent | ✗ | 1.85x | 9x |
| ReSample (Song et al., 2024) | Projection | Latent | ✓[5] | 29.5x | 8.95x |
| DCS (Ours) | Hybrid | Pixel | ✓ | **1x** | **1x** |

Table 2: Description of existing solvers used for comparison. For each solver we list the type (as described in Section 2.2), optimization space (pixel or latent), whether it requires backpropagation through a neural function evaluation (NFE, i.e., the score network call), as well as runtime and memory footprint.

We find that **DCS** either outperforms, or is comparable to all existing methods. While some methods have strong points and fail to recover the signal at other times, **DCS** is relatively consistent across these experiments. For example, **DCS** is one of few methods that has reasonable results on Motion Deblurring. DDNM, on the other hand, is very powerful across inpainting tasks in general, but fails to perform Motion Deblurring and has underwhelming qualitative performance on many other tasks. We note that some methods underperform in our benchmarks compared to the results in their papers': DDNM and PSLD due to the presence of noise in our benchmarks, and STSL and Resample for reasons we discuss in Appendix C.

We also notice that **DCS** provides a very significant speedup and memory footprint reduction compared to all methods, as notated in Table 2. We achieve this by not requiring backpropagation of the score network, as well as limiting the required number of neural function evaluations by using the more precise form of Tweedie's formula.

### 5.1 HIGHER NOISE LEVEL

We run identical benchmarks to the previous section, but at a higher noise level $\sigma_{\mathbf{y}} = 0.1$. We display the results for the FFHQ and ImageNet datasets in Table 1. We again see **DCS** achieve comparable or superior results at every task. Projection methods such as DDNM and DDRM further deteriorate, as they overfit and attempt to reproduce the noise. Other methods such as PSLD do not degrade as much, however we can see from qualitative examples that they are likely underfitting in all regimes, and therefore only gain noise-robustness by sacrificing performance at lower noise levels. Both **DCS** and DPS strike a much clearer balance between overfitting and underfitting, which is apparent from quantitative results as well as qualitative results in Figures 5 and 6.

### 5.2 ABLATION ON THE NOISE-AWARE MAXIMIZATION OPTIMIZER

We investigate how the choice of optimizer and parameters affects the noise-aware maximization algorithm in **DCS**. We note that the flexibility of using an optimizer enables us to make use of a frequentist stopping criterion as detailed in Section 4. In Figure 8 we run **DCS** with AdamW Loshchilov et al. (2017), SGD with momentum, and vanilla SGD to solve the SRx4 task on a subset of FFHQ. Runs of each optimizer at learning rate scaling factors are displayed to show the best performance, ensuring a fair comparison. It is clear in Figure 8 that the addition of a momentum term to the optimization process (both present in AdamW and SGD with momentum) can attain a higher level of image fidelity and solver stability than vanilla SGD. This provides empirical evidence for optimizer bias having an effect on solver performance in **DCS**. We see from this experiment that AdamW produces the most consistent results across learning rates, which motivates its use in our implementation.

---

[3]Latent-DPS is a direct application of DPS Chung et al. (2023) to latent diffusion models. It is also mentioned in (Rout et al., 2023).

## 6 CONCLUSION

We proposed an effective adjustment to the diffusion-based inverse problem solver framework in the literature that improves speed and stability. Observing that the marginals of the diffusion process which solves the inverse problem is Gaussian distributed at each time $t$, we derived a simple, single-parameter likelihood model, whose sole unknown variate may be obtained via a tractable maximum likelihood estimation algorithm. This casts a frequentist's light on the inverse problem framework in diffusion-based solvers, as opposed to the prevailing current of posterior and projection-based perspectives. We leveraged this new perspective to create a noise-aware maximization scheme, and demonstrated the effectiveness of our method via a suite of numerical experiments.

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

# A    ADDITIONAL THEOREMS AND PROOFS

## A.1    PROOFS FOR REPRODUCTION OF TWEEDIE'S APPROXIMATION

For completeness, we demonstrate necessity by including the proof for Lemma 3.1.

*Proof (of Lemma 3.1).*

$$\frac{1}{\sqrt{\alpha_t}}\left[\mathbf{x}_t + \sigma_t^2\nabla_{\mathbf{x}_t}\log p_t(\mathbf{x}_t|\mathbf{x}_0)\right] = \frac{1}{\sqrt{\alpha_t}}\left[\mathbf{x}_t - \nabla_{\mathbf{x}_t}\sigma_t^2\frac{1}{2\sigma_t^2}||\mathbf{x}_t - \sqrt{\alpha_t}\mathbf{x}_0||_2^2\right] \tag{25}$$

$$= \frac{1}{\sqrt{\alpha_t}}\left[\mathbf{x}_t - (\mathbf{x}_t - \sqrt{\alpha_t}\mathbf{x}_0)\right] \tag{26}$$

$$= \mathbf{x}_0. \tag{27}$$

$\square$

To demonstrate sufficiency, we show that the inverse of Lemma 3.1 also holds.

**Lemma A.1** (A sufficient condition for Tweedie's formula). *If $\mathbf{x}_0$ can be recovered via Eq. 17, then $p_t(\mathbf{x}_t|\mathbf{x}_0)$ takes the form Eq. 16.*

*Proof (of Lemma A.1).* Suppose that

$$\mathbf{x}_0 = \frac{1}{\sqrt{\alpha_t}}\left[\mathbf{x}_t + \sigma_t^2\nabla_{\mathbf{x}_t}\log p_t(\mathbf{x}_t|\mathbf{x}_0)\right] \tag{28}$$

Then we may re-arrange terms, obtaining

$$\frac{\sqrt{\alpha_t}\mathbf{x}_0 - \mathbf{x}_t}{\sigma_t^2} = \nabla_{\mathbf{x}_t}\log p_t(\mathbf{x}_t|\mathbf{x}_0). \tag{29}$$

Taking the anti-derivative of both sides, we conclude that

$$\log p_t(\mathbf{x}_t|\mathbf{x}_0) = \frac{1}{2\sigma_t^2}||\mathbf{x}_t - \sqrt{\alpha_t}\mathbf{x}_0||_2^2 + C. \tag{30}$$

Since $\log p_t(\mathbf{x}_t|\mathbf{x}_0)$ can only take this form when $p_t(\mathbf{x}_t|\mathbf{x}_0)$ is a simple isotropic Gaussian distribution, we conclude our proof. $\square$

*Proof (of Theorem 3.2).* Observing that Lemmas 3.1 and A.1 are converses of each other, we demonstrate that the conditions stated in Lemma 3.1 are necessary and sufficient. $\square$

## A.2    THEOREMS FOR SUFFICIENCY

We set up Theorems to show that the estimator in Eq. 21 is a sufficient statistic under different properties of $\mathcal{A}$. Letting $\mathbf{f}(\mathbf{y})$ be the function that obtains $\nabla\log p_t(\mathbf{x}_t|\mathbf{x}_0)$ via Eq. 21, we show that $\mathbf{y}$ is measurable under the sigma algebra induced by the measurement $\mathbf{f}$.

Intuitively, we demonstrate that $\mathbf{f}(\mathbf{y})$ contains as much information as possible about the underlying signal $\mathbf{x}_0$ as can be gathered via $\mathbf{y}$. The theoretical and intuitive statements can be summarized by the simple conditional equivalence

$$p(\mathbf{y}|\epsilon_{\mathbf{y}_*}) = p(\mathbf{y}|\mathbf{x}_0). \tag{31}$$

First, we consider two simple and theoretically similar cases: when $\mathbf{y} = \mathcal{A}(\mathbf{x})$ is noise-free, and when $\mathcal{A}$ is linear.

**Theorem A.2** ($\epsilon_{\mathbf{y}_*}$ is a sufficient statistic). *Let $\mathbf{y} = \mathcal{A}(\mathbf{x}_0) + \eta$ be an observation from the forward measurement model, and let*

$$\epsilon_{\mathbf{y}_*} = \arg\max_{\epsilon_{\mathbf{y}}}\log p\left(\mathbf{y}\left|\frac{1}{\sqrt{\alpha_t}}(\mathbf{x}_t + \sigma_t^2\epsilon_{\mathbf{y}_*})\right.\right). \tag{32}$$

*Then*

$$p(\mathbf{y}|\epsilon_{\mathbf{y}_*}) = p(\mathbf{y}|\mathbf{x}_0). \tag{33}$$

*given that either $\eta = 0$, or $\mathcal{A}$ is linear.*

We now investigate the general noisy case where $\mathcal{A}$ is allowed to be nonlinear. We find that our results can still be quite general: we only need to assume $\mathcal{A}$ surjective, meaning that there exists some $\mathbf{x} \in \text{domain}(\mathcal{A})$ such that $\mathcal{A}(\mathbf{x}) = \mathbf{y}$. In fact, this result is slightly stronger — we are able to show that optimality holds for $\mathcal{A}$ that are compositions of linear and surjective functions.

**Theorem A.3.** *Let $\epsilon_{\mathbf{y}_*}$ be as defined in Theorem A.2. Suppose the twice-differentiable operator $\mathcal{A} := \mathbf{P}^T \circ \phi$ is composed of $\mathbf{P} : \mathbb{R}^d \to \mathbb{R}^r$, a linear projection, and $\phi : \mathbb{R}^n \to \mathbb{R}^r$, an arbitrary surjective function. We have that*

$$p(\mathbf{y}|\epsilon_{\mathbf{y}_*}) = p(\mathbf{y}|\mathbf{x}_0). \tag{34}$$

To prove Theorems A.2 and A.3, we establish the following Lemma which characterizes useful information about $\mathbf{x}_0^*$.

**Lemma A.4.** *Suppose $\mathbf{y} \in \mathbb{R}^k$ is fixed, $\mathbf{x}_t \in \mathbb{R}^n$, with twice differentiable linear operator $\mathcal{A} : \mathbb{R}^n \to \mathbb{R}^k$. Then, for $\epsilon_{\mathbf{y}} = \nabla_{\mathbf{x}_t} \log p_t(\mathbf{x}_t|\mathbf{x}_0)$ which maximizes $p(\mathbf{y}|\mathbf{x}_0)$, the following holds true:*

1. *if $\eta = 0$ (i.e. the noiseless regime), $\mathcal{A}(\mathbf{x}_0) = \mathcal{A}(\mathbf{x}_t + \sigma_t^2 \epsilon_{\mathbf{y}}^*)$*

2. *if $\mathcal{A}$ is surjective, $\mathcal{A}(\mathbf{x}_0) = \mathcal{A}(\mathbf{x}_t + \sigma_t^2 \epsilon_{\mathbf{y}}^*)$*

3. *if $\mathcal{A}$ is linear, $\langle \mathbf{y} - \mathcal{A}(\mathbf{x}_t + \sigma_t^2 \epsilon_{\mathbf{y}}^*), \mathcal{A}(\mathbf{x}_t + \sigma_t^2 \epsilon_{\mathbf{y}}^*) - \mathcal{A}(\mathbf{x}_0) \rangle = 0$.*

An interpretation of statement 3 reads that the optimal solution $\epsilon_{\mathbf{y}}^*$ for estimating $\mathbf{x}_0$ is orthogonal to the error to $\mathbf{y}$ in the linear case. The requirements of statement 3 may be relaxed to the statement $\mathcal{A}(\mathbf{x}) - \mathcal{A}(\mathbf{z})$ is in the range of the Jacobian of $\mathcal{A}$ at $\mathbf{z}$, however this is less intuitive than linearity. We avoid invoking linearity of $\mathcal{A}$ as long as possible to illustrate the fact that other transformations may share this property as well.

*Proof (of Lemma A.4).* We will make use of the bijective mapping $\mathbf{z} \mapsto \mathbf{x}_t + \sigma_t^2 \epsilon_{\mathbf{y}}$, and charactarize the minima which maximize $\log p(\mathbf{y}|\mathbf{x}_0)$. We can solve the optimization problem,

$$\arg \min_{\mathbf{z}} \ ||\mathbf{y} - \mathcal{A}(\mathbf{z})||_2^2$$

A minima to this objective can be characterized by he first order necessary condition,

$$\nabla_{\mathbf{z}} ||\mathbf{y} - \mathcal{A}(\mathbf{z})||_2^2 = -2\mathbf{J}_{\mathbf{z}}[\mathcal{A}](\mathbf{z})^T (\mathbf{y} - \mathcal{A}(\mathbf{z}))$$
$$= -2\mathbf{J}_{\mathbf{z}}[\mathcal{A}](\mathbf{z})^T (\mathcal{A}(\mathbf{x}_0) - \eta - \mathcal{A}(\mathbf{z})) := 0.$$

We can confirm it is a minima by checking the solution of the above with,

$$\mathbf{H}_{\mathbf{z}} \left[ ||\mathbf{y} - \mathcal{A}(\mathbf{z})||_2^2 \right] (\mathbf{z}^*) = 2\nabla_{\mathbf{z}} \left[ \mathbf{J}_{\mathbf{z}}[\mathcal{A}](\mathbf{z})^T (\mathcal{A}(\mathbf{x}_0) + \eta) \right] (\mathbf{z}^*)$$
$$= 2\mathbf{J}_{\mathbf{z}}[\mathcal{A}](\mathbf{z}^*)^T \mathbf{J}_{\mathbf{z}}[\mathcal{A}](\mathbf{z}^*) + \sum_{j=1}^{k} \mathbf{H}_{\mathbf{z}}[\mathcal{A}_{(j)}](\mathbf{z}^*) (\mathbf{y} - \mathcal{A}(\mathbf{z}^*))$$
$$\succcurlyeq 0.$$

If $\eta = 0$, we have that $\mathcal{A}(\mathbf{x}_0) = \mathbf{y}$, and therefore choosing any $\mathcal{A}(\mathbf{z}^*) = \mathcal{A}(\mathbf{x}_0)$ satisfies the first order condition. The second order condition is furthermore satisfied, as $\mathbf{y} - \mathcal{A}(\mathbf{z}^*) = 0$, meaning,

$$\mathbf{H}_{\mathbf{z}} \left[ ||\mathbf{y} - \mathcal{A}(\mathbf{z})||_2^2 \right] (\mathbf{z}^*) = 2\mathbf{J}_{\mathbf{z}}[\mathcal{A}](\mathbf{z}^*)^T \mathbf{J}_{\mathbf{z}}[\mathcal{A}](\mathbf{z}^*) \succcurlyeq 0.$$

This satisfies statement 1. Statement 2 is satisfied similarly, by choosing the same $\mathbf{z}$. Note that this case differs, in that $\mathbf{z} = \mathbf{x}_0$ is no longer necessarily a valid solution.

Statement 3, is already satisfied in the cases where $\mathcal{A}$ has rank equal to the dimension of its co-domain (if $d = n$, this is equivalent to being full rank), since $\mathbf{y} - \mathcal{A}(\mathbf{z}^*) = 0$. The more interesting case is where $\mathcal{A}$ is low-rank.

To show orthogonality between $\mathbf{y} - \mathcal{A}(\mathbf{z}^*)$ and $\mathcal{A}(\mathbf{z}^*) - \mathcal{A}(\mathbf{x}_0)$ in other cases, we let $\eta = \delta + \delta_\perp$. We can choose an optimal value for $\delta_\perp$ that satisfies $\delta_\perp^* = \inf_{\delta_\perp} \{ \|\mathbf{y} - \mathcal{A}(\mathbf{z}^*) - \delta_\perp\|_2^2 \}$, for the optimal value, $\mathbf{z}^*$. Due to the non-negativity and $0$ preserving properties of norms, we have,

$$
\begin{aligned}
\delta_\perp^* &= \mathbf{y} - \mathcal{A}(\mathbf{z}^*) \\
&= \mathcal{A}(\mathbf{x}_0) + \eta - \mathcal{A}(\mathbf{z}^*) \\
&= \mathcal{A}(\mathbf{x}_0) + \delta_\perp^* + \delta^* - \mathcal{A}(\mathbf{z}^*) \\
\implies \delta^* &= \mathcal{A}(\mathbf{z}^*) - \mathcal{A}(\mathbf{x}_0).
\end{aligned}
$$

At the optima of the original objective, $\mathbf{z}^*$, the first order necessary condition dictates that,

$$
\mathbf{J}_{\mathbf{z}}[\mathcal{A}](\mathbf{z}^*)^T (y - \mathcal{A}(\mathbf{z}^*)) = \mathbf{J}_{\mathbf{z}}[\mathcal{A}](\mathbf{z}^*)^T \delta_\perp^* := 0.
$$

For a linear $\mathcal{A}$, the Jacobian is constant, so let $\mathbf{J}_{\mathbf{z}}[\mathcal{A}] = \mathbf{J}$. Therefore, $\mathbf{J}^T \delta_\perp^* = 0$, meaning $\delta_\perp^* \in \mathcal{N}(\mathbf{J}^T)$.

Simultaneously, since $\delta^* = \mathcal{A}(\mathbf{x}) - \mathcal{A}(\mathbf{z}^*) = \mathcal{A}(\mathbf{x} - \mathbf{z}^*) = \mathbf{J}(\mathbf{x} - \mathbf{z}^*)$, we have $\delta^* \in \mathcal{R}(\mathbf{J}^T)$. Therefore do to the orthogonality of range and null spaces of matrix, $\langle \delta_\perp^*, \delta^* \rangle = 0$, completing the proof.

$\square$

We are now able to prove the theorems in the main text.

*Proof of Theorem A.2.* We leverage the theory of sufficient statistics to demonstrate our result. Namely, if $\epsilon_{\mathbf{y}_*}$ is a sufficient statistic for $\mathbf{y}$, then,

$$
p(\mathbf{y}|\epsilon_{\mathbf{y}_*}) = p(\mathbf{y}|\epsilon_{\mathbf{y}_*}, \mathbf{x}_0) = p(\mathbf{y}|\mathbf{x}_0). \tag{35}
$$

Therefore it suffices to demonstrate that $\epsilon_{\mathbf{y}_*}$ is a sufficient statistic for $\mathbf{y}$.

By the Neyman-Fisher Factorization theorem, we have that a necessary and sufficient condition is if there exists non-negative functions $g_\theta$ and $h$ such that

$$
p(\mathbf{y}|\mathbf{x}_0) = g(\epsilon_{\mathbf{y}_*}, \mathbf{x}_0) h(\mathbf{y}). \tag{36}
$$

We observe that since $\eta \sim \mathcal{N}(\mathbf{0}, \sigma_{\mathbf{y}}^2 \mathbf{I})$, our random variable $\mathbf{y}$ can be characterized by the density function

$$
p(\mathbf{y}|\mathbf{x}_0) = \mathcal{N}(\mathbf{y}; \mu = \mathcal{A}(\mathbf{x}_0), \Sigma = \sigma_{\mathbf{y}}^2 \mathbf{I}). \tag{37}
$$

Therefore, letting $\mathbf{y}_{\epsilon_{\mathbf{y}_*}} = \mathcal{A}(\frac{1}{\sqrt{\alpha_t}} (\mathbf{x}_t + \sigma_t^2 \epsilon_{\mathbf{y}_*}))$, we can write

$$
p(\mathbf{y}|\mathbf{x}_0) = (2\pi\sigma_{\mathbf{y}}^2)^{-n/2} \exp\left( -\frac{1}{2\sigma_{\mathbf{y}}^2} \|\mathbf{y} - \mathcal{A}(\mathbf{x}_0)\|_2^2 \right) \tag{38}
$$

$$
= (2\pi\sigma_{\mathbf{y}}^2)^{-n/2} \exp\left( -\frac{1}{2\sigma_{\mathbf{y}}^2} \left( \|\mathbf{y} - \mathbf{y}_{\epsilon_{\mathbf{y}_*}}\|_2^2 + \|\mathbf{y}_{\epsilon_{\mathbf{y}_*}} - \mathcal{A}(\mathbf{x}_0)\|_2^2 + 2\langle \mathbf{y} - \mathbf{y}_{\epsilon_{\mathbf{y}_*}}, \mathbf{y}_{\epsilon_{\mathbf{y}_*}} - \mathcal{A}(\mathbf{x}_0) \rangle \right) \right) \tag{39}
$$

$$
= (2\pi\sigma_{\mathbf{y}}^2)^{-n/2} \exp\left( -\frac{1}{2\sigma_{\mathbf{y}}^2} \|\mathbf{y}_{\epsilon_{\mathbf{y}_*}} - \mathcal{A}(\mathbf{x}_0)\|_2^2 \right) \exp\left( -\frac{1}{2\sigma_{\mathbf{y}}^2} \|\mathbf{y} - \mathbf{y}_{\epsilon_{\mathbf{y}_*}}\|_2^2 \right), \tag{40}
$$

where the third equality is due to Lemma A.4. In the case that $\mathcal{A}$ is surjective, or the noiseless regime, statements 2 and 1 respectively satisfy the equality above trivially, as $\mathbf{y} = \mathbf{y}_{\epsilon_{\mathbf{y}_*}}$. If the operator is otherwise linear, statement 3 shows the cross term vanishes.

Therefore, we can assign

$$g(\epsilon_{\mathbf{y}_*}, \mathbf{x}_0) = (2\pi\sigma_{\mathbf{y}}^2)^{-n/2} \exp\left( \frac{1}{2\sigma_{\mathbf{y}}^2} ||\mathbf{y}_{\epsilon_{\mathbf{y}_*}} - \mathcal{A}(\mathbf{x}_0)||_2^2 \right) \tag{41}$$

$$h(\mathbf{y}) = \exp\left( \frac{1}{2\sigma_{\mathbf{y}}^2} ||\mathbf{y} - \mathbf{y}_{\epsilon_{\mathbf{y}_*}}||_2^2 \right). \tag{42}$$

In the case where the measurement process $\mathcal{A}(\mathbf{x}) = \mathbf{y}$ is noiseless, this implies $h(\mathbf{y}) = 1$. $\qquad\square$

We now modify the argument in order to relax the linearity assumption.

*Proof of Theorem A.3.* Let $\mathbf{z} = \frac{1}{\sqrt{\alpha_t}}\left( \mathbf{x}_t + \sigma_t^2 \epsilon_{\mathbf{y}} \right)$, and $\mathbf{z}^* = \arg\min_{\mathbf{z}} \{||\mathbf{y} - \mathcal{A}(\mathbf{z})||\}$.

Since $\mathbf{z}^*$ minimizes the objective $||\mathbf{y} - \mathcal{A}(\mathbf{z})||$, we also have that,

$$\phi(\mathbf{z}^*) := \arg\min_{\alpha} \{||\mathbf{y} - \mathbf{P}^T(\alpha)||\} = \arg\max_{\alpha} \; p(\mathbf{y}|\alpha).$$

We can invoke Lemma A.4 to say

$$||\mathbf{y} - \mathbf{P}^T\phi(\mathbf{x}_0)||_2^2 = ||\mathbf{y} - \mathbf{P}^T\phi(\mathbf{z}^*)||_2^2 + ||\mathbf{P}^T\phi(\mathbf{z}^*) - \mathbf{P}^T\phi(\mathbf{x}_0)||_2^2,$$

since $\mathbf{P}^T$ is a linear operator, and $\phi(\mathbf{z}^*)$ satisfies the conditions in the lemma. Therefore, we have,

$$p(\mathbf{y}|\mathbf{x}_0) = (2\pi\sigma_{\mathbf{y}}^2)^{-n/2} \exp\left( -\frac{1}{2\sigma_{\mathbf{y}}^2} ||\mathbf{y} - \mathbf{P}^T\phi(\mathbf{x}_0)||_2^2 \right)$$

$$= (2\pi\sigma_{\mathbf{y}}^2)^{-n/2} \exp\left( -\frac{1}{2\sigma_{\mathbf{y}}^2} ||\mathbf{y} - \mathbf{P}^T\phi(\mathbf{z}^*)||_2^2 \right) \exp\left( -\frac{1}{2\sigma_{\mathbf{y}}^2} ||\mathbf{P}^T\phi(\mathbf{z}^*) - \mathbf{P}^T\phi(\mathbf{x}_0)||_2^2 \right).$$

We assign terms,

$$g(\mathbf{z}_*, \mathbf{x}_0) = (2\pi\sigma_{\mathbf{y}}^2)^{-n/2} \exp\left( -\frac{1}{2\sigma_{\mathbf{y}}^2} ||\mathbf{P}^T\phi(\mathbf{z}^*) - \mathbf{P}^T\phi(\mathbf{x}_0)||_2^2 \right) \tag{43}$$

$$h(\mathbf{y}) = \exp\left( -\frac{1}{2\sigma_{\mathbf{y}}^2} ||\mathbf{y} - \mathbf{P}^T\phi(\mathbf{z}^*)||_2^2 \right), \tag{44}$$

$$\tag{45}$$

and once again invoke the Neyman-Fisher Factorization theorem to show $\mathbf{z}^*$ is sufficient for $\mathbf{y}$. Since $\epsilon_{\mathbf{y}_*}$ is a bijective mapping from $\mathbf{z}^*$, we have that $\epsilon_{\mathbf{y}_*}$ is sufficient, and similarly to Theorem A.2 we state, $p(\mathbf{y}|\epsilon_{\mathbf{y}_*}) = p(\mathbf{y}|\epsilon_{\mathbf{y}_*}, \mathbf{x}_0) = p(\mathbf{y}|\mathbf{x}_0)$. $\qquad\square$

Figure 9: A demonstration of our solver, **DCS**, solving two inverse problems on natural images from the CelebA-HQ dataset. Motion blur (left), and box dropout (right) are examples of forward operators that are non-invertible. We show further results in Section 5

# B  ADDITIONAL EXPERIMENTS

## B.1  FURTHER NOISE EXPERIMENTS

| FFHQ | SR ×4 | | | Random Inpainting | | | Box Inpainting | | | Gaussian Deblurring | | | Motion Deblurring | | |
|---|---|---|---|---|---|---|---|---|---|---|---|---|---|---|---|
| | LPIPS ↓ | PSNR ↑ | FID ↓ | LPIPS ↓ | PSNR ↑ | FID ↓ | LPIPS ↓ | PSNR ↑ | FID ↓ | LPIPS ↓ | PSNR ↑ | FID ↓ | LPIPS ↓ | PSNR ↑ | FID ↓ |
| Ours | 0.2287 | 20.362 | 100.94 | 0.2067 | 22.999 | 89.312 | 0.2005 | 21.298 | 40.099 | 0.2109 | 24.009 | 82.132 | 0.2301 | 22.306 | 90.403 |
| DPS | 0.2000 | 22.588 | 92.791 | 0.2290 | 22.808 | 90.739 | 0.2118 | 20.278 | 81.491 | 0.2268 | 25.020 | 83.686 | 0.2479 | 20.767 | 91.972 |
| DDNM | 0.7812 | 9.8324 | 387.43 | 0.8721 | 15.573 | 233.15 | 0.9966 | 12.607 | 287.79 | 1.4475 | 3.5686 | 408.85 | 1.3328 | 3.1782 | 393.24 |
| ReSample | 0.5704 | 19.948 | 179.35 | 0.6892 | 20.014 | 200.06 | 0.4958 | 17.530 | 160.47 | 0.5409 | 21.166 | 162.40 | 0.6380 | 19.875 | 194.69 |

Table 3: Quantitative experiments on FFHQ 256x256-1K at $\sigma_{\mathbf{y}} = 0.5$. We compare against pixel-based solvers (upper half) and latent-based solvers (lower half).

## B.2  SUBSET OF FFHQ USED IN OTHER WORKS

| | SR ×4 | | | Random Inpainting | | | Box Inpainting | | | Gaussian Deblurring | | | Motion Deblurring | | | Cost | |
|---|---|---|---|---|---|---|---|---|---|---|---|---|---|---|---|---|---|
| | LPIPS ↓ | PSNR ↑ | SSIM ↑ | LPIPS ↓ | PSNR ↑ | SSIM ↑ | LPIPS ↓ | PSNR ↑ | SSIM ↑ | LPIPS ↓ | PSNR ↑ | SSIM ↑ | LPIPS ↓ | PSNR ↑ | SSIM ↑ | Time ↓ | Mem. ↓ |
| Ours | **0.074** | **29.51** | **0.811** | **0.052** | **31.13** | **0.850** | **0.102** | **22.07** | 0.761 | **0.078** | **29.92** | **0.817** | **0.051** | **32.32** | **0.833** | 1x | 1x |
| DPS | 0.132 | 27.10 | 0.729 | 0.084 | 30.91 | 0.833 | 0.107 | 21.62 | 0.755 | 0.090 | 28.26 | 0.767 | 0.108 | 26.816 | 0.726 | 6x | 3.2x |
| MCG | 0.112 | 27.07 | 0.784 | 0.877 | 11.02 | 0.02 | 0.905 | 10.883 | 0.001 | 0.176 | 24.89 | 0.768 | - | - | - | 6.1x | 3.2x |
| DDNM | 0.242 | 27.63 | 0.587 | 0.230 | 27.92 | 0.604 | 0.194 | 23.08 | 0.639 | 0.287 | 27.24 | 0.561 | 0.642 | 8.682 | 0.165 | 1.75x | 1x |
| Latent-DPS | 0.324 | 20.086 | 0.473 | 0.249 | 22.64 | 0.570 | 0.227 | 22.184 | 0.595 | 0.209 | 23.512 | 0.600 | 0.217 | 22.930 | 0.582 | 6.1x | 8.9x |
| PSLD | 0.311 | 20.547 | 0.491 | 0.250 | 22.84 | 0.579 | 0.221 | 22.23 | 0.607 | 0.200 | 23.77 | 0.614 | 0.213 | 23.277 | 0.596 | 7.5x | 15x |
| STSL | 0.242 | 27.63 | 0.587 | 0.230 | 27.92 | 0.604 | 0.194 | 23.08 | 0.639 | 0.287 | 27.24 | 0.561 | 0.641 | 10.17 | 0.245 | 1.85x | 9x |
| ReSample | 0.090 | 29.024 | 0.791 | 0.053 | 30.99 | 0.844 | 0.156 | 20.71 | **0.778** | 0.113 | 29.19 | 0.784 | 0.197 | 27.65 | 0.706 | 29.5x | 8.95x |

Table 4: Quantitative evaluation of our method on FFHQ 256x256, following the experimental setup of (Song et al., 2024). We compare against pixel-based solvers (upper half) and latent-based solvers (lower half).

# C  IMPLEMENTATION DETAILS

We provide implementation details of our experiments, as well as those for other experiments we compare against.

## C.1  OUR METHOD

Our proposed DCS has just two primary hyperparameters, as described in the table below. First is the number of time steps $T$. This has relatively little effect on our model performance on most tasks. However, it is occasionally helpful to increase $T$, especially in box inpainting, where there is zero signal from $\mathbf{y}$ in the masked region. Here, higher $T$ allows the diffusion model to obtain a better solution in this unconditional diffusion process. Second, we have the choice of `minimizer`, which is by default the Adam optimizer Kingma and Ba (2014). However, in the case of linear $\mathcal{A}$, this optimizer can be replaced by the closed form analytical solution to $\mathcal{A}(\mathbf{x}) = \mathbf{y}$.

For nearly all experiments, we use the Adam optimizer with 50 optimization steps and a learning rate of 1. The exceptions are the random inpainting and box inpainting tasks, where there is no conditioning information on the masked pixels. This requires more denoising steps, as the diffusion process is totally unconditional inside the mask, up to local correlations learned inside the score

network $s_\theta$. Here, we use the analytical solver with $\mathcal{A}^\dagger = \mathcal{A}$. Similarly, for nearly all experiments we use $T = 50$ as found in Table 7, with the exception being random inpainting and box inpainting tasks, where we found that taking $T = 1000$ steps improved performance. However, there is little increase in runtime, since the minimization step is much faster here.

| Notation | Definition |
|----------|------------|
| $T$ | The number of diffusion steps used in the sampler. |
| minimizer | The minimizer used to solve for $\epsilon_\mathbf{y}$. |

## C.2   LATENT MODELS ON IMAGENET

We note that previous latent models use the pretrained weights in (Rombach et al., 2022) for $256 \times 256$ resolution datasets. However, there are no published weights in the GitHub repository for unconditional ImageNet, making a fair comparison of our method against latent models more involved. To this end, we leverage a significantly more powerful Stable Diffusion v1.5 model, with publicly available weights on HuggingFace for our experiments. The measurements and the output images are appropriately scaled for a fair comparison.

## C.3   STSL

At the time of writing this work, we did not find publicly available code for STSL (Rout et al., 2024). Therefore, we implement the algorithm ourselves in our codebase, and use the hyperparameters provided in the paper.

## C.4   RESAMPLE

We directly use the published code of ReSample (Song et al., 2024) with no changes in our paper. We discuss two notable aspects of the experiments with ReSample. First, the implementation on GitHub differs from that pseudocode discussed in the paper. Namely, the pseudocode in the paper describes enforcing latent- and pixel-based consistency occasionally during an otherwise unconditional sampling process.

In the code we observed that the sampling step taken is actually a DPS (Chung et al., 2022a) sampling step, which includes a posterior-based guidance step that takes an expensive gradient of the noise function. To see this, note that L255 in the `resample_sampling` function in `ddim.py` calls a function `measurement_cond_fn`, which is defined at L62 in `main.py` and passed into the resampling function. This function is a member of the class `PosteriorSampling` defined in L53 in `condition_methods.py`. Inspecting this class, we note that it calls `torch.autograd.grad` on the diffusion step as a function of `x_prev` (L33 or L39). In other words, a gradient is computed for the measurement norm with respect to the input to the diffusion model, i.e., a DPS step.

We closely investigated this DPS step in our experiments, ultimately concluding that it has a significant effect on the performance of the algorithm, and that it was a *more* fair comparison to include this step, rather than removing it. However, the inclusion of this sampling step has two primary effects. First, it results in further increases the computation time of ReSample. Second it reveals that ReSample relies significantly on a posterior-based formulation, applying additional resampling steps at each stage.

In experiments, we note that ReSample is significantly slower than other algorithms during sampling (see Table 2). For example, sampling $\sim 1000$ images with ImageNet takes more than two weeks on an A6000 GPU. Since we run five different experimental conditions for each dataset, this was an unacceptably long runtime for our academic resources. Therefore, we reduce the number of diffusion steps $T$ of ReSample in our experiments, from 500 reported in (Song et al., 2024) to 50. However, we do provide a single experiment from the (Song et al., 2024) paper, where we reproduce the hyperparameters and dataset (a 100 image subset of FFHQ). We note that (Song et al., 2024) took a subset of the FFHQ dataset, where performance differed from the full $256\times256$-1K dataset performance (c.f. Table 1). Since the subset was not published, we selected a dataset based where ReSample obtained the same performance with its default parameters in (Song et al., 2024) (Table 4).

### C.5 DDRM

We used the version of DDRM which is implemented in the DDNM codebase. While DDRM may theoretically be able to handle deblurring tasks, due to the high rank of the forward operators, the SVD cannot be explicitly defined in memory, and no existing code-base for DDRM supplies fast and memory-saving versions of these operators. Because of the relatively poor performance of DDRM compared to DDNM, and the fact that DDRM can be considered a subtype of DDNM (see Appendix of Wang et al. (2022)), we do not run on deblurring tasks.

## D  FURTHER QUALITATIVE COMPARISONS

We provide further qualitative examples from the FFHQ 256×256-1K and ImageNet 256×256-1K datasets accompanying our quantitative evaluation in Table 1.

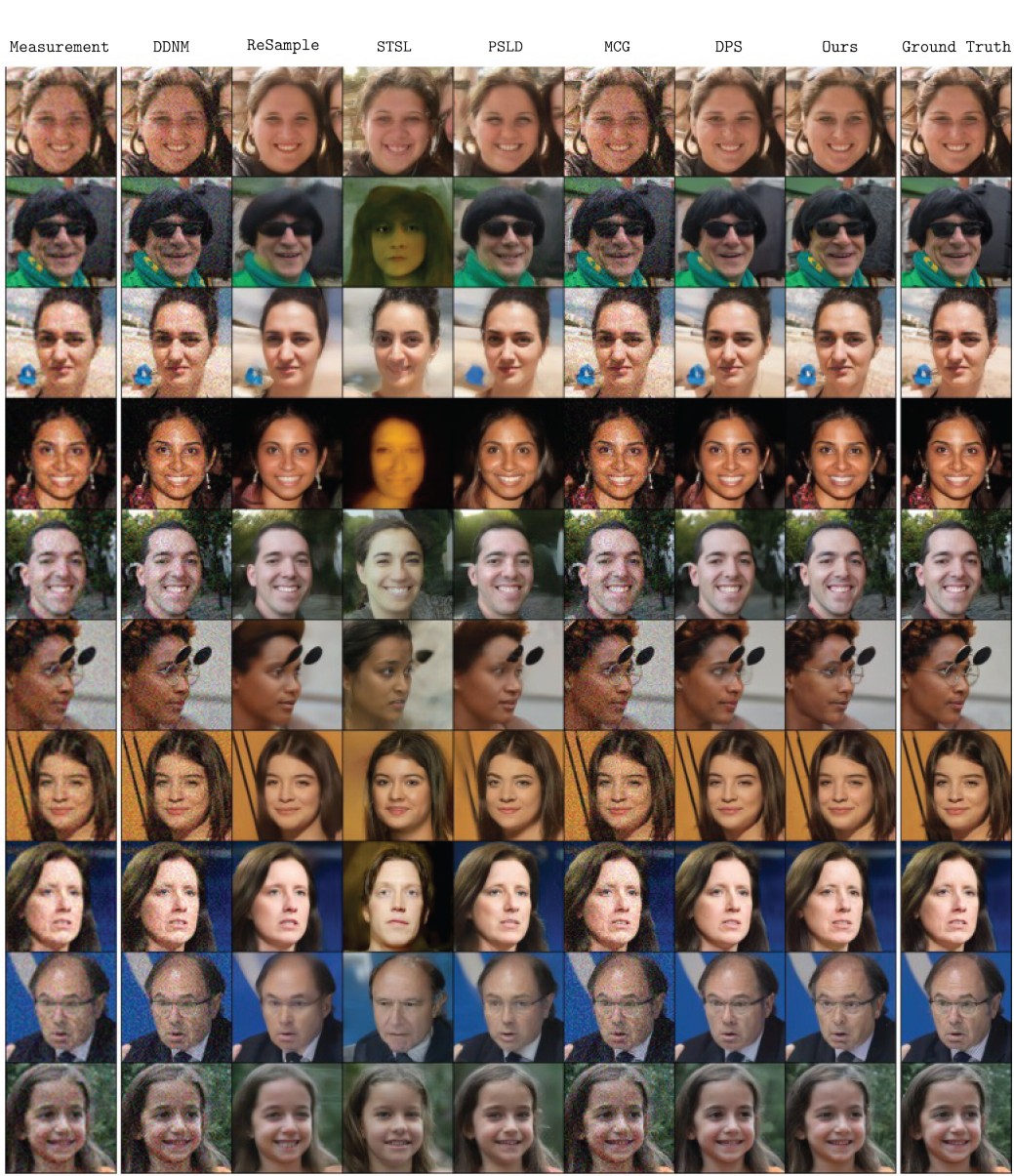

Figure 10: Comparison against competing works on FFHQ 256×256-1K dataset with the 4× super-resolution task.

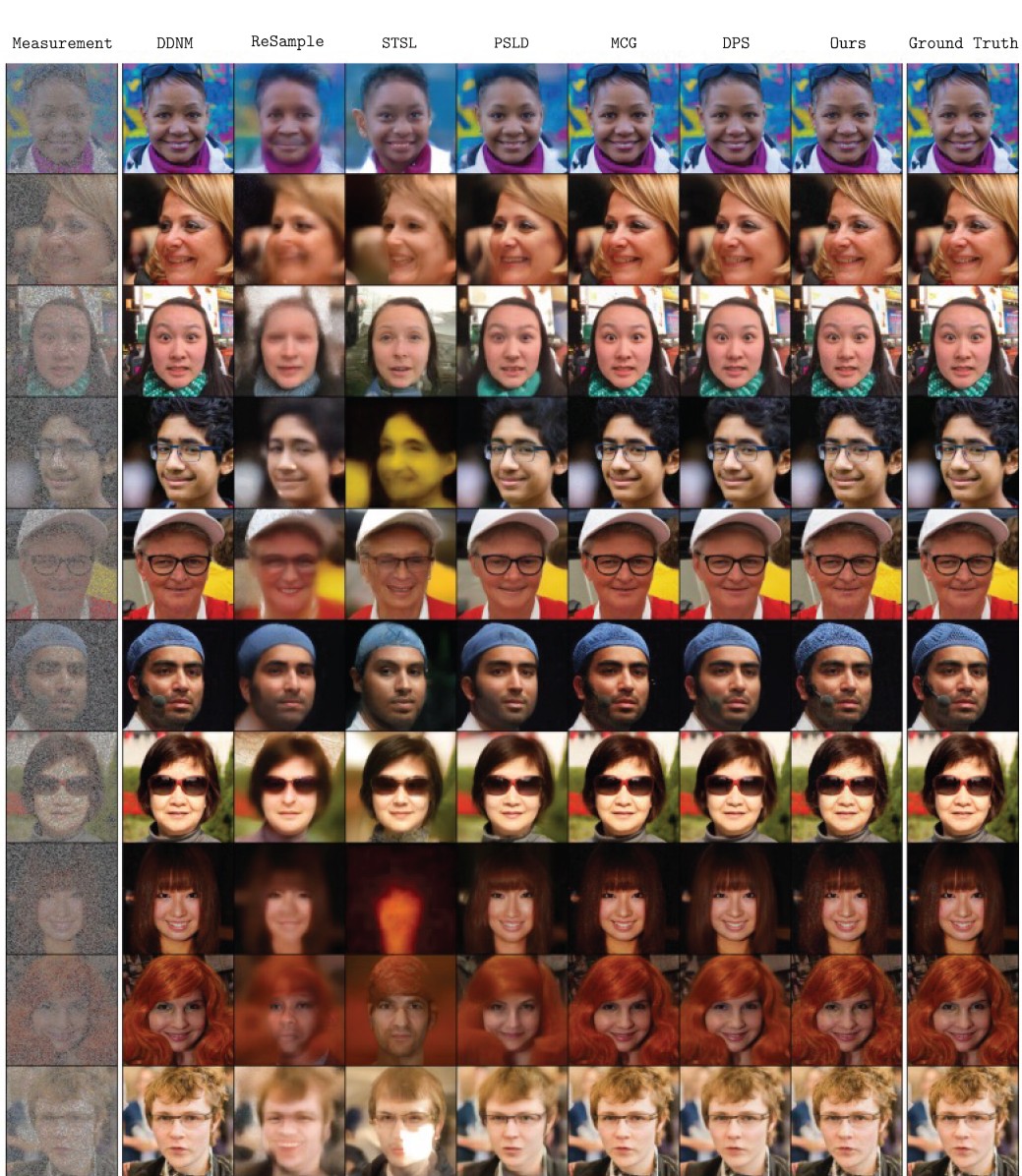

Figure 11: Comparison against competing works on FFHQ 256×256-1K dataset with the random inpainting task.

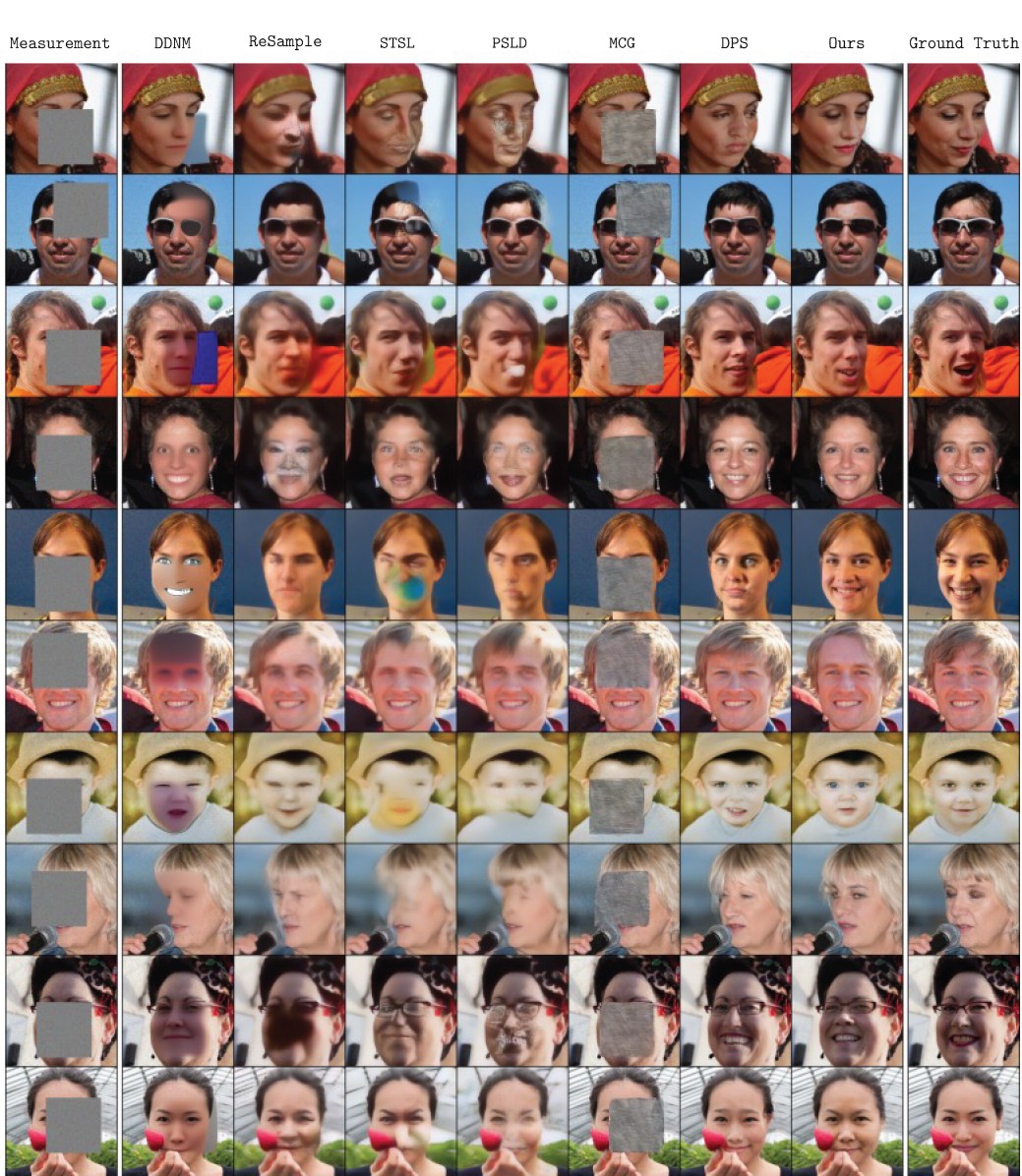

Figure 12: Comparison against competing works on FFHQ 256×256-1K dataset with the box inpainting task.

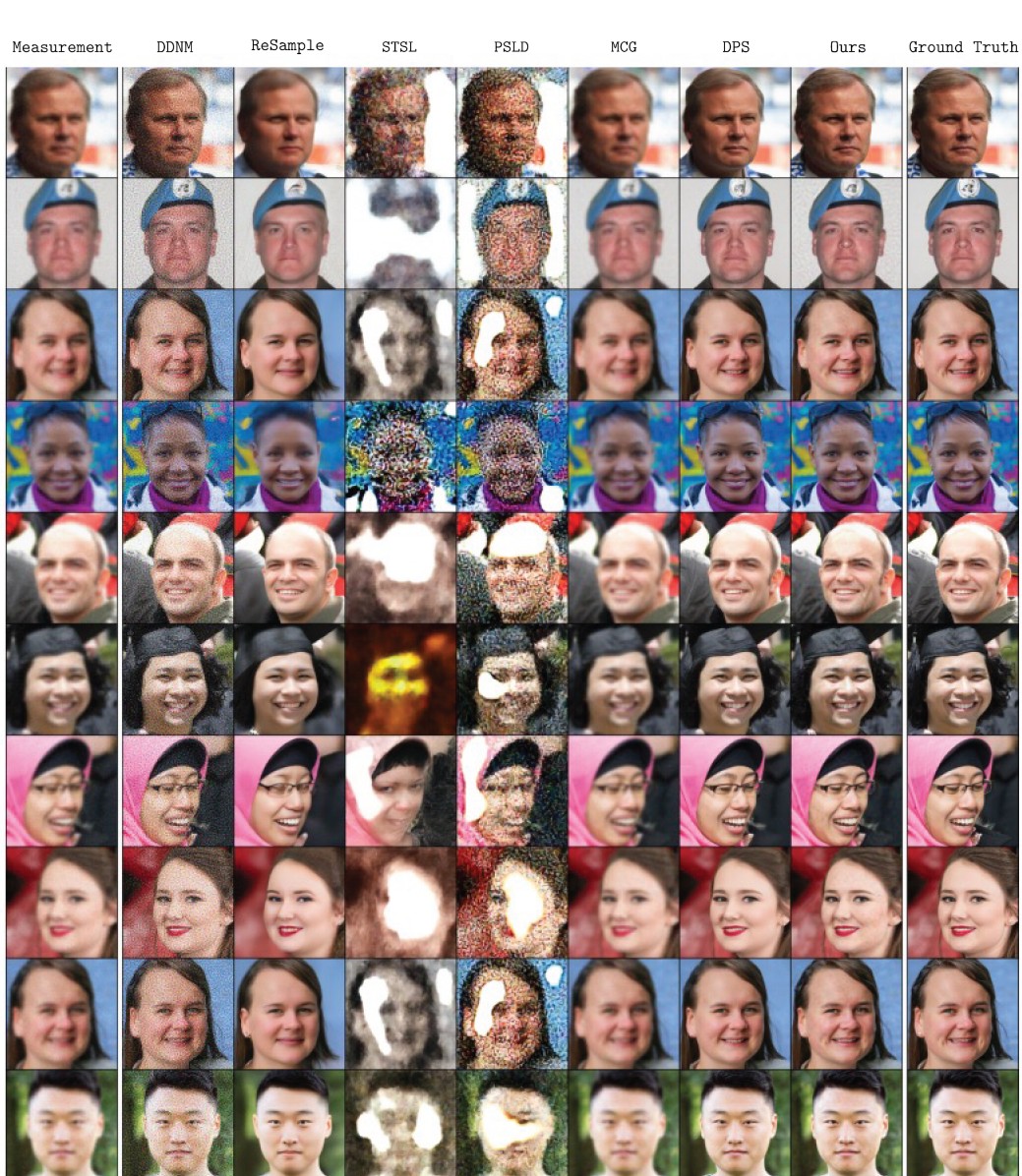

Figure 13: Comparison against competing works on FFHQ 256×256-1K dataset with the Gaussian deblurring task.

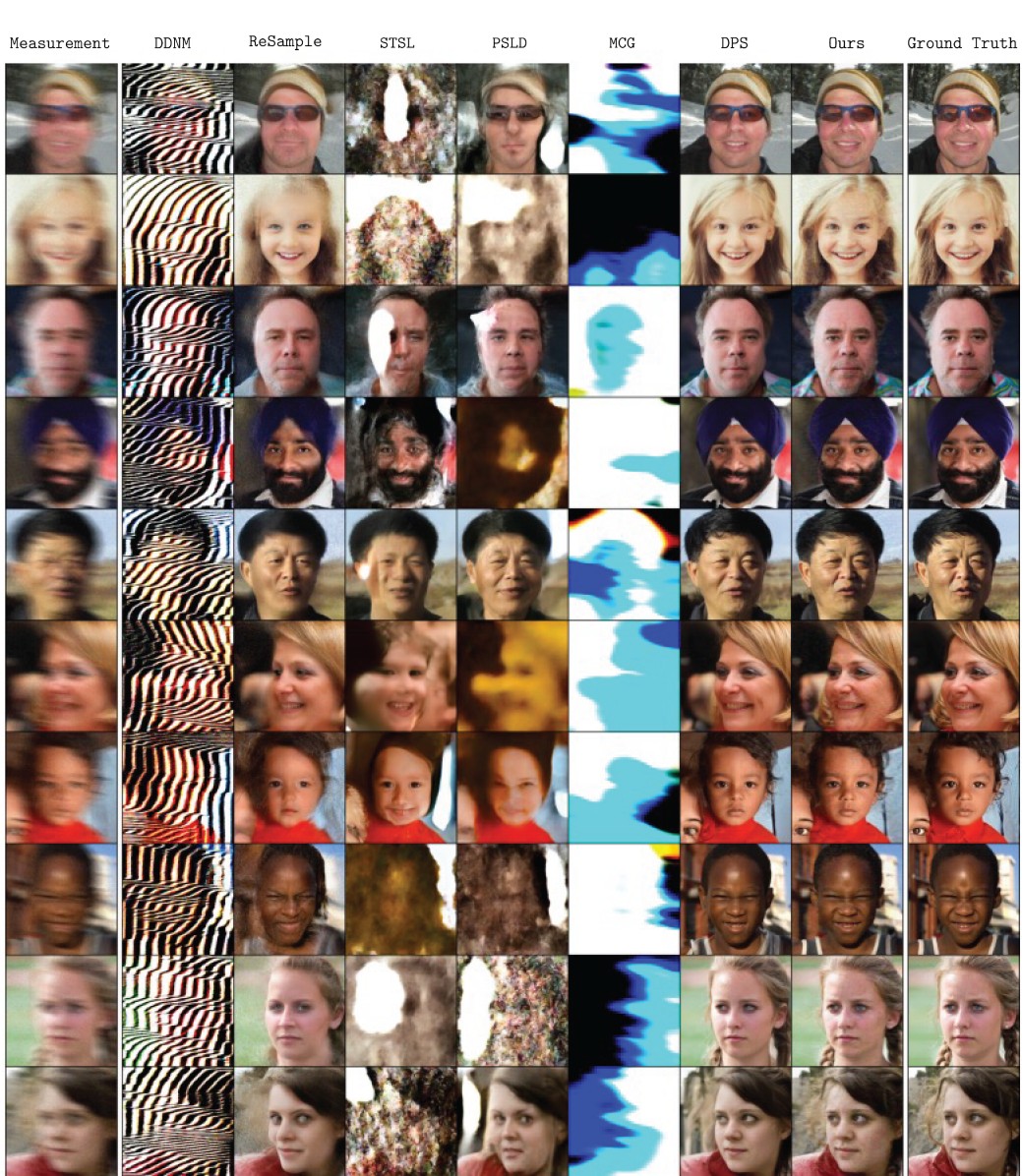

Figure 14: Comparison against competing works on FFHQ 256×256-1K dataset with the motion deblurring task.

Measurement  DDNM  ReSample  STSL  PSLD  MCG  DPS  Ours  Ground Truth

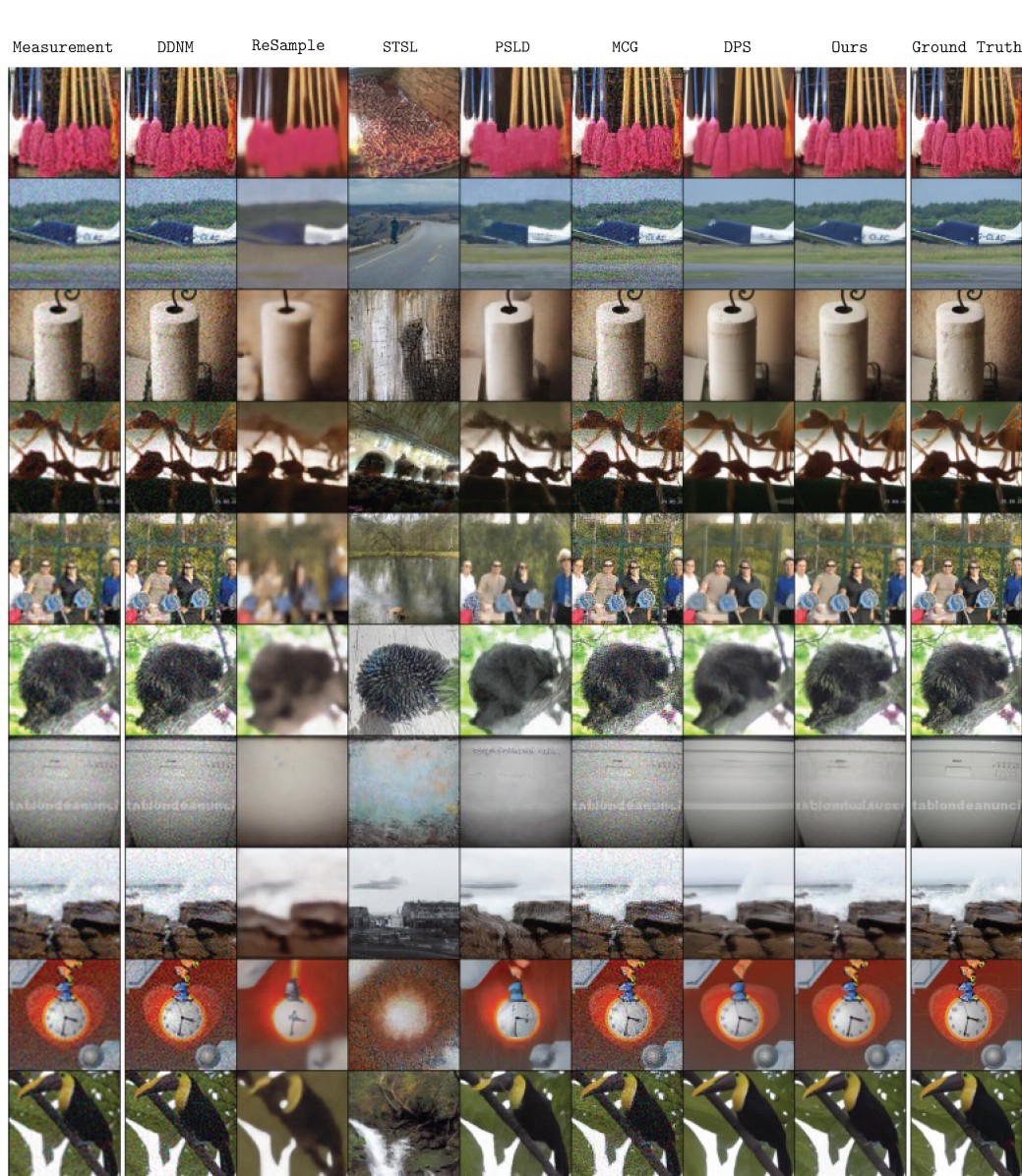

Figure 15: Comparison against competing works on FFHQ 256×256-1K dataset with the 4× super-resolution task.

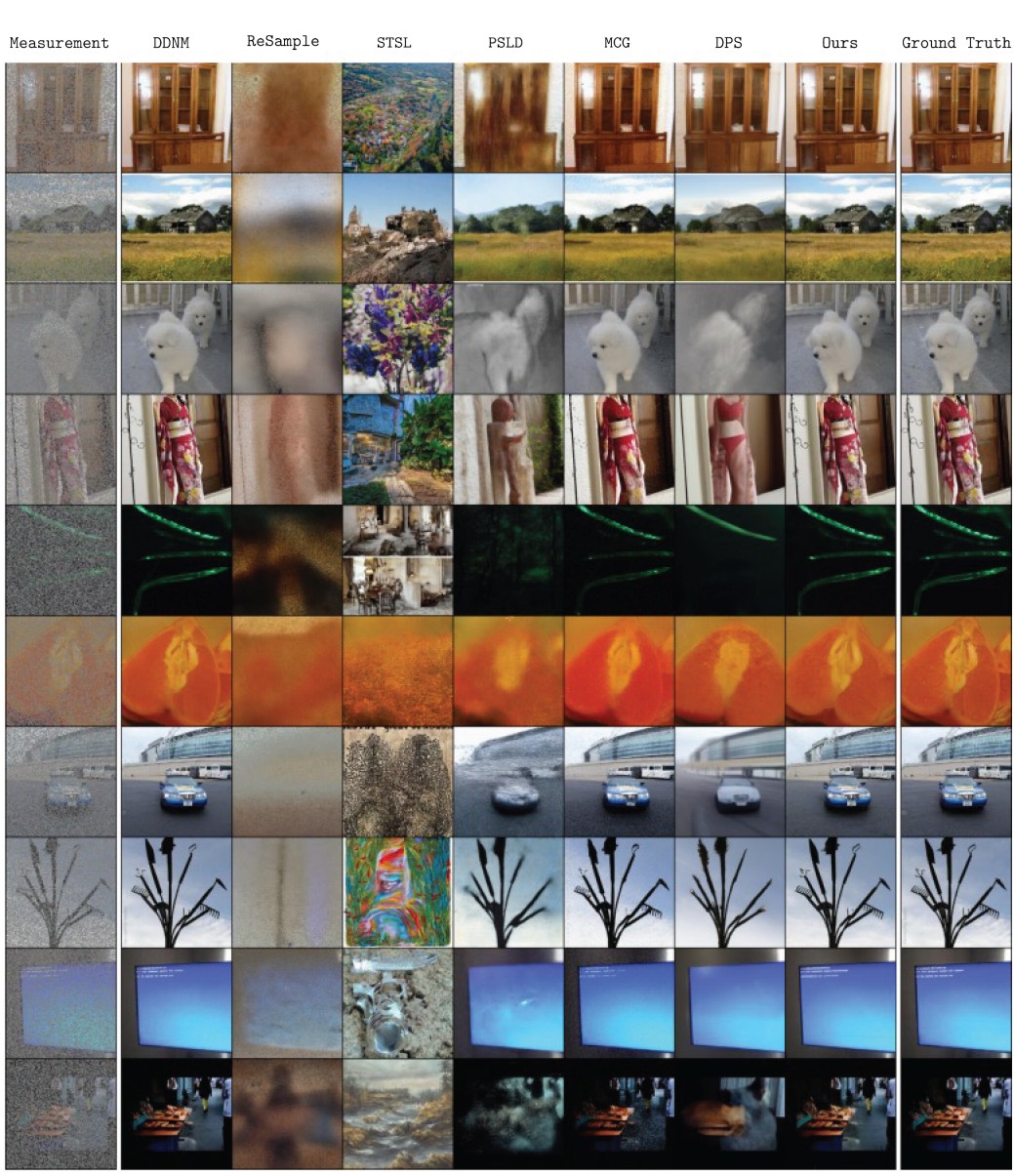

Figure 16: Comparison against competing works on ImageNet 256×256-1K dataset with the random inpainting task.

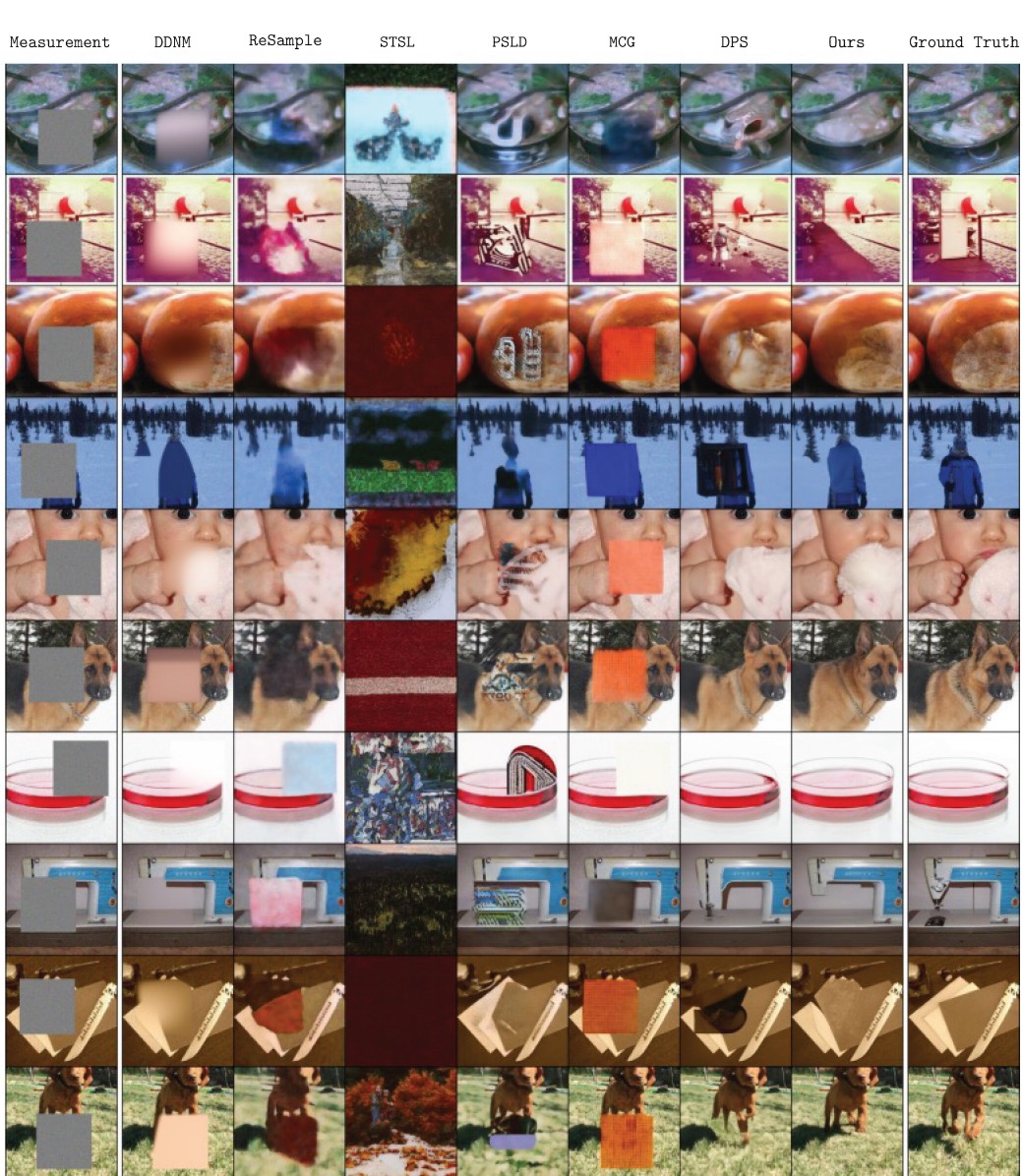

Figure 17: Comparison against competing works on FFHQ 256×256-1K dataset with the box inpainting task.

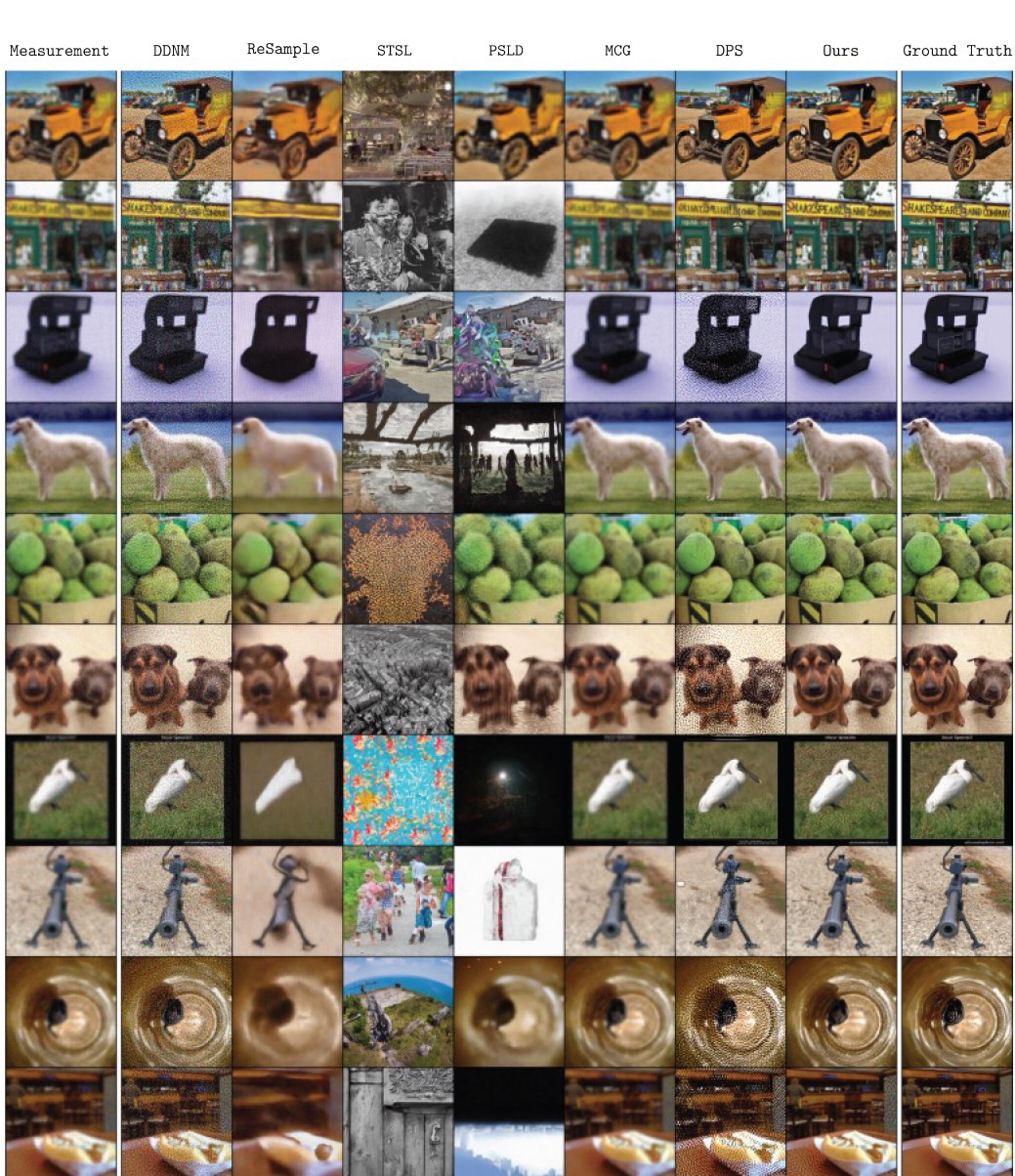

Figure 18: Comparison against competing works on ImageNet 256×256-1K dataset with the Gaussian deblurring task.

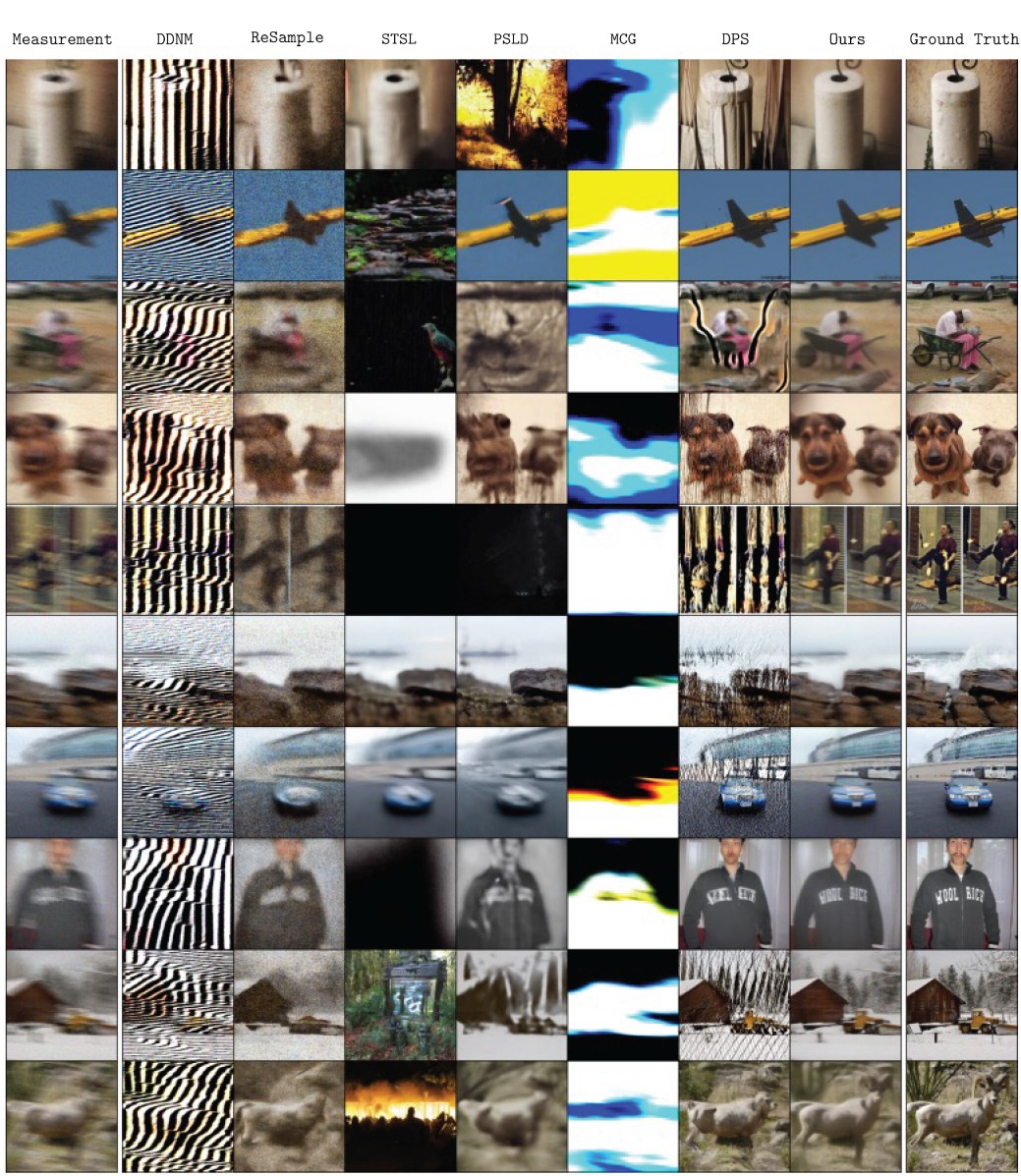

Figure 19: Comparison against competing works on ImageNet 256×256-1K dataset with the motion deblurring task.

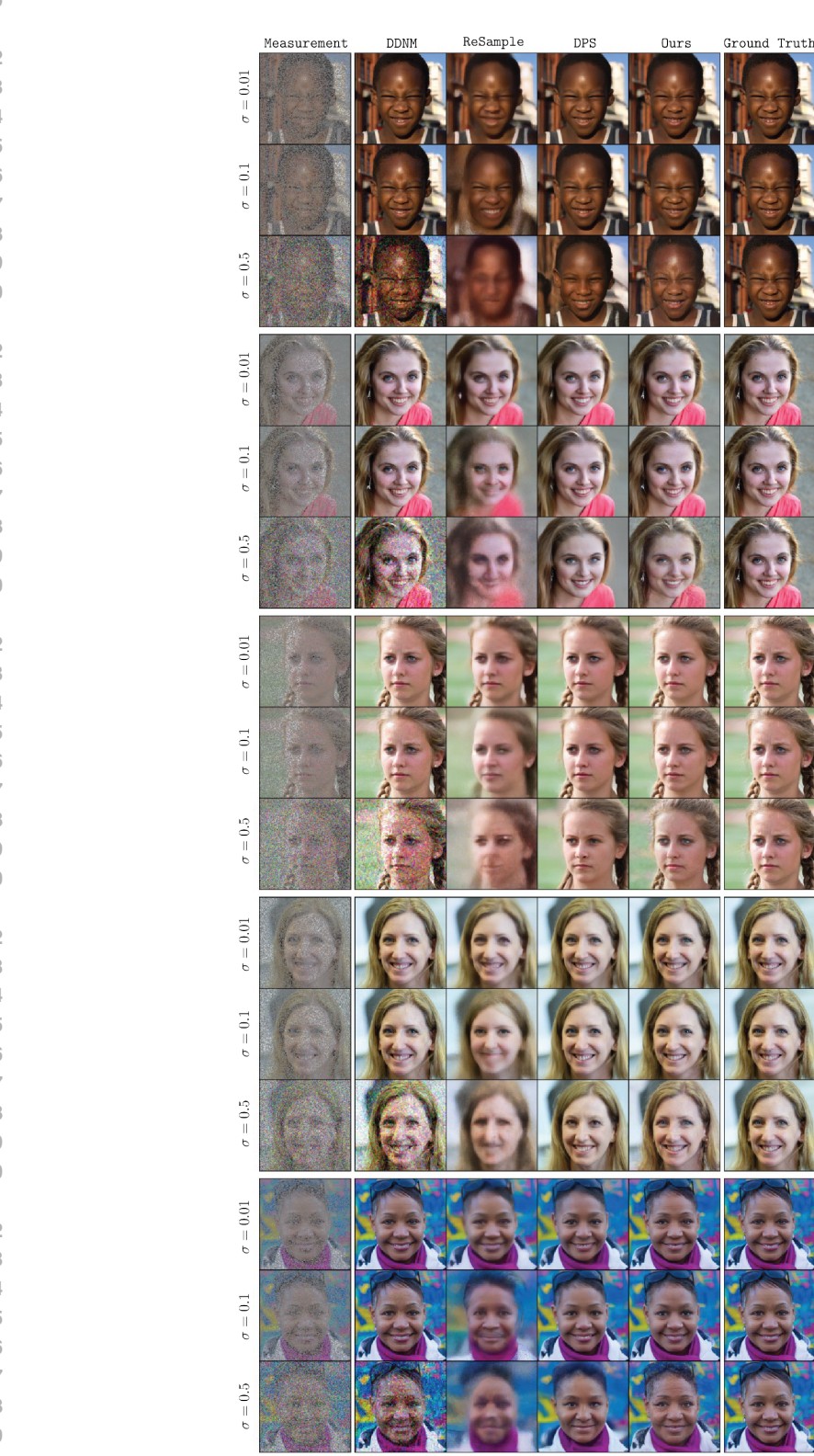

Figure 20: Comparison against competing works on FFHQ 256×256-1K dataset with the random inpainting task at various noise levels.

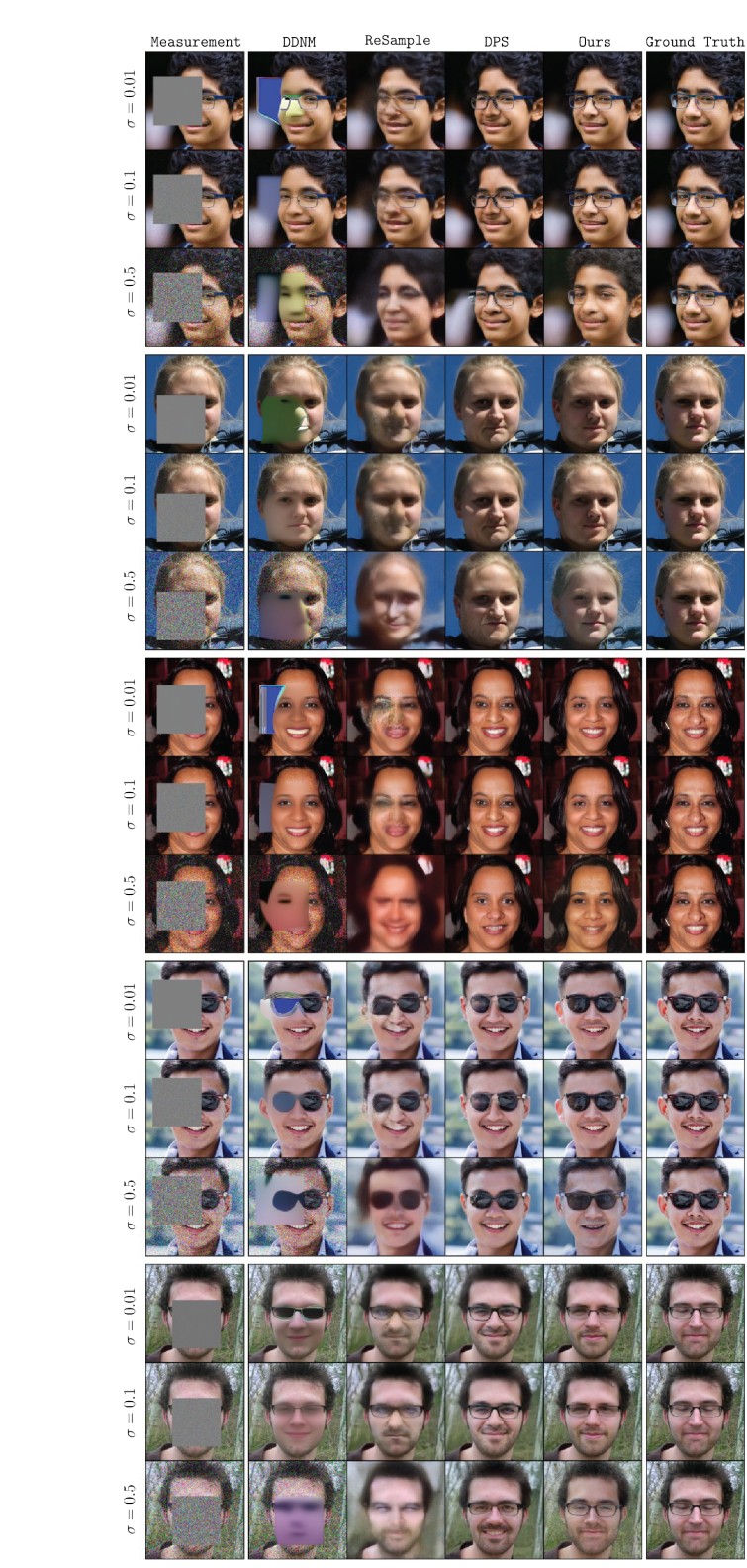

Figure 21: Comparison against competing works on FFHQ 256×256-1K dataset with the box inpainting task at various noise levels.

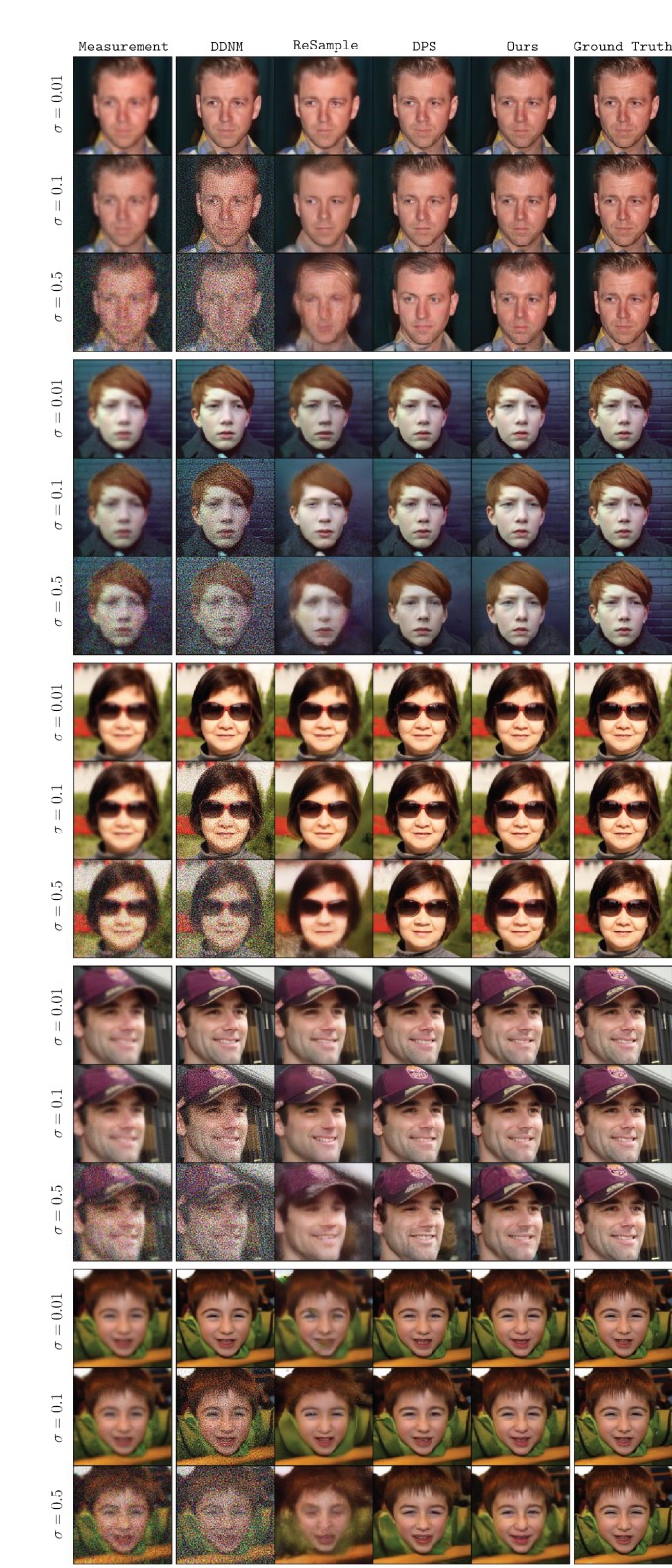

Figure 22: Comparison against competing works on FFHQ 256×256-1K dataset with the Gaussian deblurring task at various noise levels.

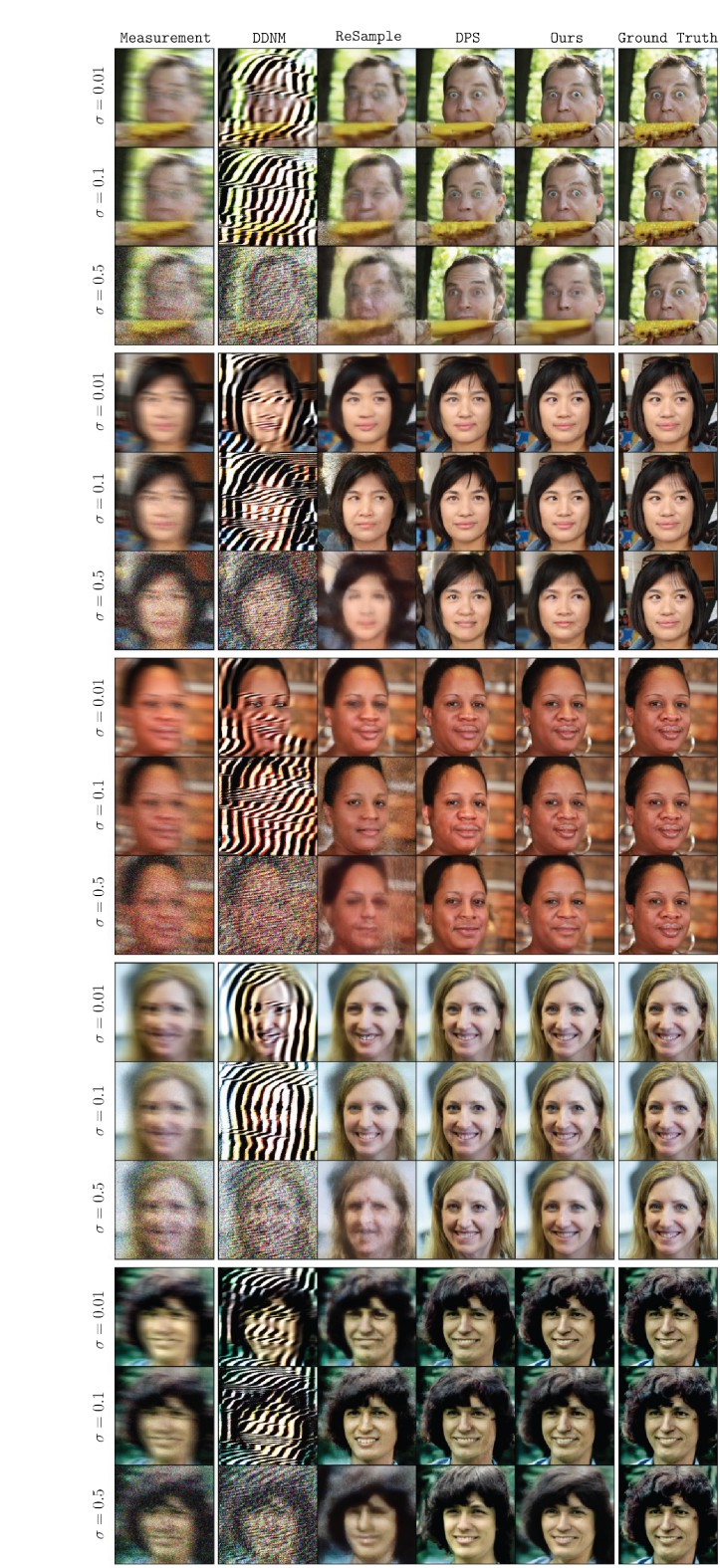

Figure 23: Comparison against competing works on FFHQ 256×256-1K dataset with the motion deblurring task at various noise levels.

