# OpenReview forum: "Fast and Noise-Robust Diffusion Solvers for Inverse Problems: A Frequentist Approach"
_ICLR.cc/2025/Conference — Submitted to ICLR 2025_

### Official Review · Reviewer_Cmd7 · 2024-11-01

**Soundness:** 4
**Presentation:** 4
**Contribution:** 4
**Rating:** 8
**Confidence:** 4

**Summary:**

The paper investigates solving inverse problems with score-based diffusion models. Particularly, they address the problem that comes with noisy data and the inconsistency between generated image and noisy measurement common algorithms encounter. To this end, the authors propose a novel algorithm that overcomes this problem. The algorithm is theoretically analyzed and extensively investigated numerically on various tasks and noise-levels.

**Strengths:**

I really like this paper. The idea is to the best of my knowledge novel and solves a very important problem in the field. The clarity of the argument is excellent as is the presentation. The numerical results are very comprehensive giving a transparent assessment of the potential of the method for common computer vision tasks. I also really like that they test their method on high-noise levels showing that it may indeed have practical relevance, even for severely ill-posed inverse problems (which they haven't tested on).

**Weaknesses:**

In my view there are no important weaknesses.

The writing is at times a little bit off:
- Introduction: "Generally, A is assumed to be non-invertible, meaning that any solution x satisfying A(x) = y ...". This statement is false. Many inverse problems are invertible: Gaussian and motion deblurring, X-ray tomography. The former is even considered in this paper. This then has consequences for the argument that follows.
- line 50: they refer to "smoothness" as a function, similar to total variation. I guess they mean the squared H1-seminorm ||\grad u||_2^2?
- line 68: "While already effective, this approach suffers from a unique problem where the explicit form of the consistency error ||A(x) − y|| only exists for x = x0 (Chung et al., 2022a)." The error of course exists everywhere but cannot be easily or readily evaluated at the correct location.
- line 214: "Of course, this strategy is only correct when two conditions simultaneously hold true: (1) the measurement operator A is linear, and (2) the inverse problem is noiseless, i.e, η is identically 0." I don't follow their argument. I would be happy to say that (1) and (2) are sufficient for this to be correct (but may not be necessary in general).
- line 260: "Inverse problem solvers (Section 2.2) face a fundamental ..." This of course is only true for score-based diffusion method. Most algorithms for inverse problems do not have this problem.

**Questions:**

I am fully sold on this paper. Of course the authors may want to reply to the comments as described above.

---

> ### Author Response · Authors · 2024-11-18
>
> We would like to express our gratitude that the reviewer appreciates both the theoretical and empirical results of our work, and for bringing several insightful shortcomings of our work to our attention. Below, we address the reviewer’s concerns on a point-by-point basis.
>
> **Introduction: Many ill-conditioned problems are invertible (such as Gaussian & motion blur):** We agree that in theory some operators we test in this work (such as Gaussian and motion blur) are perfectly invertible. However, due to ill-conditioning, finite precision, and bounded image sizes operators like Gaussian and motion blur are not invertible in a real-world setting.
>
> Indeed, this point is worth noting, and we have changed “Generally” to “In many cases”, and added this detail as a footnote to the sentence.
>
> **Line 50: Smoothness is not defined explicitly** We acknowledge that we do not specify what type of smoothness we refer to; we purposefully leave this term open-ended as a myriad of approaches exist using different functionals to evaluate smoothness, Lipschitz continuity being among those.
>
> **Line 214:** Indeed, these conditions are better characterized as necessary rather than sufficient conditions, and have replaced “only correct when…” with “often restricted to situations where..” to remove this implication.
>
> **Line 260:**  We agree that this problem is only present in diffusion-based solvers and we have added this qualifier.

---

> > ### Comment · Reviewer_Cmd7 · 2024-11-18
> >
> > I agree with all except one of the responses. Even in a real-world setting (as they described), the Gaussian convolution is still invertible. Of course, this does not imply that we want or should invert it (due to ill-conditioning, noise etc).

---

> > > ### Author Response · Authors · 2024-11-18
> > >
> > > We thank the reviewer for their comment. Indeed, the Gaussian convolution is invertible. However, when applied to bounded images (as in many real-world settings, and in our work), the Gaussian blur operator is not simply a Gaussian convolution. Rather, it is the composition of a Gaussian convolution and a projection that sets the pixels outside the original image bounds to zero. This projection is not invertible, which renders the entire operation non-invertible. We have further clarified this in footnote 1 of our updated manuscript.

---

> > > > ### Comment · Reviewer_Cmd7 · 2024-11-19
> > > >
> > > > I feel like this discussion is leading nowhere. Here is my final comment on the topic: The convolution G itself is invertible both in theory and practice. Apparently the authors have performed intensity clipping without mentioning it, e.g. C(x) = min(max(x,0), 1) and set Ax = C(G(x)). Of course, C is not invertible (both in theory and practice) but also nondifferentiable and nonlinear.

---

> > > > > ### Author Response · Authors · 2024-11-19
> > > > >
> > > > > We are in total agreement with the reviewer on the nature of the Gaussian convolution.
> > > > >
> > > > > However, we wish to rectify a miscommunication in our comments -- we do not compose the Gaussian convolution with a clipping operator. We compose the Gaussian convolution with padding and projection operators that handle the borders of the image, as is standard practice for Gaussian blurs in our benchmarks (see, e.g., the implementation in DPS). These are linear but not necessarily invertible. We apologize for the confusion.

---

### Official Review · Reviewer_B1H3 · 2024-11-03

**Soundness:** 2
**Presentation:** 3
**Contribution:** 2
**Rating:** 5
**Confidence:** 4

**Summary:**

The authors propose a novel frequentist's approach to score-based diffusion inverse solvers by directly sampling with a data-conditional score. Each diffusion step can be seen as the maximum likelihood solution to a single-parameter conditional likelihood model. This model is derived through an adjusted application of Tweedie’s formula to the forward measurement model. It allows for a noise-aware maximization scheme.

**Strengths:**

The approach is scalable, and allows for a noise-aware maximization scheme with a likelihood-based stopping criterion that promotes the noise-adapted fit given knowledge of the measurement noise. The authors extensively demonstrate the performance on a variety of tasks from inverse problems.

**Weaknesses:**

1. The paper makes strong assumptions about the measurement operator $A$. Theorem A.3 assumes $A$ can be decomposed into a linear projection and a surjective function, which, while covering quite many operators, is still restrictive.

2. Although Diffusion Conditional Sampling (DCS) claims computational efficiency by avoiding score function gradients, it depends on a noise-aware maximization (NAM) scheme that adds its own complexity. The paper lacks a thorough analysis of NAM’s computational complexity.

3. The ablation study in Section 5.2 indicates that DCS’s performance depends on the optimizer used in NAM, introducing a sensitive hyperparameter. This sensitivity may reduce the method's computational advantages.

4. Despite recognizing limitations in Tweedie’s formula, the paper relies on it to estimate $x_0$, arguing that the Gaussian assumption in the reverse process justifies this. However, this justification hinges on the accuracy of the score estimate from NAM; any inaccuracies could propagate through Tweedie’s formula and affect reconstruction quality.

5. While Theorem A.3 affirms the sufficiency of the score estimate, I am not sure if the paper provides theoretical guarantees on DCS’s convergence or overall accuracy.

6. Qualitative comparisons in Figures 10-19 reveal subtle differences between DCS and other methods, often with minimal resemblance to the ground truth. This makes it hard to assert DCS’s advantage in image quality or fidelity.

7. DCS’s frequentist approach may limit its use in applications needing uncertainty quantification. Figures 10-19 show none of the methods fully recover the ground truth, but some existing methods enable posterior sampling to quantify uncertainty - a capability that DCS seems to lack.

8. It would have been beneficial to elaborate on a comparison with Y. Sun, Z. Wu, Y. Chen, B. T. Feng, and K. L. Bouman. Provable probabilistic imaging using score-based generative priors.

**Questions:**

Could the authors clarify Theorem A.3 and its proof? for example, concerning guarantees on the DCS's convergence.

Could the authors comment on point 4. under perceived Weaknesses?

---

> ### Author Response · Authors · 2024-11-18
>
> We thank the reviewer for appreciating the noise-adapted aspect of our work, and for their insightful discussion. We respond to comments in detail, on a point-by-point basis below.
>
> **Strong assumptions on the measurement operator $\mathcal{A}$ in Theorem A.3.**
> While the assumptions in A.3 do not cover all possible measurement operators, we note that all operators considered in our work (super-resolution, random inpainting, box inpainting, motion deblurring, random deblurring), are covered by Theorem A.3. We do acknowledge the possibility of a stronger theorem, and leave a generalization of Theorem A.3 to future work.
>
> **The proposed noise-aware maximization (NAM) scheme lacks computational complexity analysis.**
> NAM is an early-stopped gradient descent algorithm where the loss function is the log-likelihood (Eq. 19). Since Eq. 19 is a simple quadratic function of the optimized parameter (the conditional score), the computational complexity of NAM is upper-bounded by that of gradient descent, which is well-known to have a guaranteed linear convergence rate on general convex functions [1]. Therefore, our algorithm has at most linear complexity. In practice, this loop takes between 0 to 50 iterations.
>
> We agree that this is an important discussion, and have added it to the section on NAM.
>
> **DCS’s performance depends on the optimizer in NAM, introducing a sensitive parameter.**
> While NAM is sensitive to the optimizer, we would not consider it a tunable parameter to the end-user. Figure 7 clearly shows that AdamW exhibits stable and superior performance to all other optimizers. In fact, all our numerical experiments are conducted with AdamW, and we did not find it necessary to use any other optimizer.
>
> In fact, we would argue that our algorithm removes certain hyperparameters compared to existing approaches. For example, we do away with the scale parameter in DPS [2] and the variance schedule and early stopping parameters in ReSample [3].
>
> **The paper ultimately uses Tweedie’s formula in spite of its limitations.**
> We emphasize that we do not discourage use of Tweedie’s formula. We discourage its use *under the wrong conditions*. In diffusion models and diffusion-based inverse solvers, Tweedie’s formula is often used to estimate $x_0$. This approach is entirely valid when using the conditional score function $\nabla \log p_t(x_t | x_0)$ rather than $\nabla \log p_t(x_t)$ (Theorem 3.2). However, existing works do not make this distinction.
>
> **Moreover, NAM only estimates the score, and this could introduce error.**
> The reviewer’s concern is understandable here. Indeed, NAM is an estimator, and like any estimator, NAM only approximates the true score. (Though we remind the reviewer that all relevant scores in diffusion modeling are approximate at inference-time.) In practice, we find this error to be small (or at least smaller than the alternatives) due to the gain in performance of DCS over existing works. We leave the reduction of this estimation error to future work.
>
> **Theorem A.3 does not provide convergence or accuracy guarantees.**
> Indeed, Theorem A.3 does not provide theoretical guarantees on DCS’s convergence or overall accuracy. However, the log-likelihood (Eq. 19) optimized by DCS is a simple quadratic function of the score. Due to the strict convexity of this objective, classical linear convergence guarantees with gradient descent can be applied [1].
> Moreover, we note that statements on the convergence or accuracy of DCS are not very useful for the noisy inverse problems studied in this work: as seen in Figure 4, converging to the solution would cause overfitting. Therefore, we have refrained from making theoretical statements to this effect.
>
> **Qualitative comparisons reveal subtle differences between DCS and other methods.**
> Indeed, the differences may appear subtle to the reviewer, but DCS produces reconstructions that are state-of-the-art according to widely accepted quantitative measures (LPIPS, PSNR, and FID) used in existing work. We emphasize that Figures 10-19 are randomly selected and *not* cherry-picked.
>
> Furthermore, we would like to remind the reviewer that some tasks such as box-inpainting are fundamentally ill-posed. Therefore, ground truth recovery is simply not possible, and not a meaningful indicator of performance.
>
> **DCS approach may limit its use in applications needing uncertainty quantification.**
> While DCS employs a “frequentist approach”, we emphasize that it is still able to estimate $\nabla \log p(x_0 | y)$ which is the same quantity DPS and other posterior sampling methods estimate.
>
> We leave a detailed analysis of the uncertainty quantification of DCS to future work.

---

> > ### Author Response · Authors · 2024-11-18
> >
> > **Comparison to [4].**
> > We thank the reviewer for bringing up this work, and have added it to our discussion. It appears that [4] is closely related to DPS, as the basic reverse diffusion step can be written as $P(x_k) = g(x_k) - \alpha S_k$ (c.f. Eq. 12 in our Section 2.2 and Eq. 5 in the DPS paper [2]).
> >
> > Ultimately, [4] still relies on heavy use of the uncorrected Tweedie’s estimate, as it leverages the same uncorrected error estimate that is used by existing works for  (e.g., DDNM, DPS, etc.) in g(x) (Eq. 3), which is used in both Algorithms 1 and 2 in their work. Therefore, we would argue that their work would benefit from the corrected score estimate we propose.
> >
> >
> > [1] Convex Optimization. https://web.stanford.edu/~boyd/cvxbook/
> >
> > [2] Diffusion Posterior Sampling for General Noisy Inverse Problems. https://arxiv.org/pdf/2209.14687
> >
> > [3] Solving Inverse Problems with Latent Diffusion Models via Hard Data Consistency. https://arxiv.org/abs/2307.08123
> >
> > [4] Provable probabilistic imaging using score-based generative priors. https://arxiv.org/abs/2310.10835

---

### Official Review · Reviewer_bebv · 2024-11-04

**Soundness:** 1
**Presentation:** 2
**Contribution:** 2
**Rating:** 3
**Confidence:** 5

**Summary:**

The paper presents a likelihood-based algorithm for solving inverse problems using diffusion models as regularizers.
The approach involves solving a sequence of maximum likelihood problems with an adaptive early stopping criterion, followed by a backward diffusion step.
In the maximum likelihood stage, the authors apply Tweedie’s formula to estimate a clean sample and then regress over its corresponding residual.
To prevent overfitting the noisy measurement, the authors employ an early stopping criterion based on hypothesis testing.
The authors validate their method through extensive experiments on linear inverse problems with both pixel-space and latent-space diffusion models across different noise levels.
Also, they conduct ablation studies to assess the algorithm's sensitivity to hyperparameter settings.

**Strengths:**

- Extensive experimental suite
- Discussion of the disparity between the paper and the released code in ReSample algorithm

**Weaknesses:**

**Technical concerns**

- The paper present problematic discussion of Tweedie's formula, particularly in Lemma 3.1 and Theorem 3.2.
In diffusion models, Tweedie’s formula is valid because the transition kernels are Gaussian; see [1], Section 2.3, for a detailed proof of

$$
E(X_0 | X_t) = \frac{1}{\sqrt{\alpha_t}} (x_t + (1 - \alpha_t) \nabla \log p_t(X_t))
$$

Hence conditioning on $x_0$ and restating Tweedie's formula is irrelevant.

- Equation (17) raises concerns as $x_0$ appears on both sides of the equation.
As $x_0$ is inferred from $x_t$, it can never be accessed directly; only an estimate of its expected value is obtainable.

- The claim in lines 303–304, is problematic. The transition kernel $p_{t|0}(x_t | x_0)$ is an isotropic Gaussian for any $t$, namely

$$
p_{t|0}(x_t | x_0) = N(\sqrt{\alpha}_t x_0, (1 - \alpha_t) I)
$$

hence when t is very close to zero, the kernel is almost a dirac around $x_0$

**Mistakes**

The paper contains several substantial errors that affect the technical clarity and accuracy of the proposed methodology:

- In Equations (5) and (7), $x_t$ is omitted from the drift term of the SDE, which is necessary for a correct formulation of Diffusion Models; see Equation 11 in [2]
- In line 106 (footnote), there is an inconsistency: the score and epsilon terms are swapped. The correct expression should be $\epsilon_\theta = -\sigma_t s_\theta$
- Equation (9) does not align with the sampling scheme proposed in DDPM [3]. In DDPM, sampling is performed by recursively applying the bridge kernel $q(x_{t-1} | x_t, x_0)$; however, here the authors apply the forward kernel $p_{t|0}(x_t | x_0)$ which differs fundamentally.
- In Line 5 of Algorithm 2, the gradient should be taken with respect to $\log p_t$
- In Line 400, the correct term should refer to the Jacobian of the score, not the gradient of the log of the score.


**Irrelevant comparisons in experiments**

The experimental comparisons presented in the paper has inconsistences leading to potentially misleading conclusions:

- In Table 1, the authors compare multiple algorithms that utilize different types of priors (pixel space models and latent space models). However, changing the prior (the regularizer) change the problem being solved. Specifically, Latent DPS, PSLD, and ReSample employ latent diffusion priors, while other algorithms use pixel-space models.
Noteworthy, the use of latent diffusion models introduces significant nonlinearities into the inverse problem due to the auto-encoder.
- In the experiments, it is unclear whether the results for the authors' algorithm are based on a latent-space or pixel-space diffusion model.
This ambiguity also applies to the results reported in Tables 3 and 4 in the appendix

---

.. [1] Meng, C., Song, Y., Li, W., & Ermon, S. (2021). Estimating high order gradients of the data distribution by denoising. Advances in Neural Information Processing Systems, 34, 25359-25369.

.. [2] Song, Y., Sohl-Dickstein, J., Kingma, D. P., Kumar, A., Ermon, S., & Poole, B. (2020). Score-based generative modeling through stochastic differential equations. arXiv preprint arXiv:2011.13456.

.. [3] Ho, Jonathan, Ajay Jain, and Pieter Abbeel. "Denoising diffusion probabilistic models." Advances in neural information processing systems 33 (2020): 6840-6851.

**Questions:**

I find using "hypothesis testing" for the early stopping criterion misleading.
Hypothesis testing traditionally assesses the statistical significance of a hypothesis based on multiple samples, whereas, in the current setup, only one sample of the residual is available at each iteration of the likelihood minimization.
This make hypothesis testing less irrelevant.
Additionally, setting $\sigma_t$ as the critical probability $p_{critic}$ lacks sufficient justification.
While $\sigma_t$ values remain in [0, 1] in this setup and can thus be interpreted as probabilities, this approach may not extend to Variance Exploding (VE) diffusion models, where $\sigma$ takes values beyond the interval [0, 1].

---

> ### Author Response · Authors · 2024-11-18
>
> We thank the reviewer for their close reading of our work, and rigorous critiques. We hope to clarify some of the points of the paper, and alleviate the reviewer's concerns below in a point by point response.
>
> **Problematic discussion of Tweedie’s formula.**
>
> We provide a point-by-point response to the concerns below:
>
> **Tweedie’s formula is valid because the transition kernels are Gaussian, hence conditioning on $x_0$ and restating Tweedie’s formula is irrelevant.** We absolutely agree with the reviewer and [1] that Tweedie’s formula
> $$
> E(x_0|x_t) = \frac{1}{\sqrt{\alpha_t}}(x_t - (1 - \alpha_t) \nabla \log p_t(x_t))
> $$
> is valid for computing the posterior mean $E(x_0 | x_t)$.
>
> However, errors arise when $E(x_0 | x_t)$ is used as an approximation for $x_0$, which is a common replacement in diffusion model literature (e.g., DDIM and related deterministic diffusion samplers), as well as all existing diffusion-based inverse solvers.
>
> Our intention in Lemma 3.1 and Theorem 3.2 is to say that
> $$
> x_0 = \frac{1}{\sqrt{\alpha_t}}(x_t - (1 - \alpha_t) \nabla \log p_t(x_t))
> $$
> often does not hold. However, this point was not clear. Therefore, we have adjusted Lemma 3.1 from “Tweedie’s formula for diffusion models” to “Approximating $x_0$ with Tweedie’s formula if and only if…”, and Theorem 3.2 from “Tweedie’s formula holds” to “Tweedie’s formula predicts $x_0$ if and only if…”.
>
> Ultimately, the critical distinction is using $E[x_0 | x_t]$ (general diffusion models, DPS, DDNM, etc.) vs $x_0$ (ours) in the inverse solver. The latter is clearly a much higher quality estimate, since it is directly relevant to the inverse problem at hand ($y = \mathcal{A}(x_0) + \text{noise}$). This contributes to the improved performance of DCS compared to existing works.
>
> **Eq. 17 raises concerns as $x_0$ appears on both sides of the equation. $x_0$ cannot be accessed directly, so we must use $E[x_0|x_t]$.** The reviewer would be entirely correct in the case where only the unconditional score  $\nabla \log p_t (x_t)$ is known, and no other information is present. Of course, this is precisely the approach in standard reverse diffusion problems, and that assumed in DPS, DDNM, DDRM, etc. However, in inverse problems, we have access to more information, i.e., that stored in $y$.
>
> This motivates our maximum likelihood estimation framework. By iteratively solving for the conditional score given the model Eq. 19 and data $y$, we obtain progressively improved estimates of $x_0$.
>
> **The claim in lines 303-304 is problematic.** We thank the reviewer for bringing attention to this line. Indeed, this is a typo, as $p(x_t|x_0)$ was supposed to be $p(x_t)$ and has been corrected. The reviewer is absolutely correct that at $t$ close to 0, $p_t(x_t|x_0)$ is a Gaussian around $x_0$.
>
> **Typos.** We thank the reviewer for catching these errors, and have corrected them in the updated manuscript. We agree that this improves the clarity and accuracy of our proposal.
>
> **In Table 1, latent-space models are compared with pixel-space models, which change the problem being solved.** We respectfully disagree. While the hurdles latent-space solvers and pixel-space solvers face are different, they ultimately solve the same inverse problem, which is an ultimate goal of our work. Moreover, it is standard to compare between the two classes of algorithms (see PSLD, STSL, ReSample). Additionally, even among pixel-space models, we obtain state-of-the-art results.
>
> **It is unclear whether the results are based on a latent-space or pixel-space diffusion model.** Our algorithm and results are based on pixel space diffusion only (Table 2).
>
>
> **Using “hypothesis testing” for the early stopping criterion [is] misleading.** We emphasize that this is simply comparison, and that we do not claim to perform true hypothesis testing, only a test inspired by it (Lines 356-357). We ultimately guided the design of our test via empirical performance results.
>
> **Hypothesis testing typically uses multiple samples, whereas the proposed setup has one.** Our setup does in fact assume that we have $d$ samples, where $d$ is the dimensionality of the image. We have clarified this in the updated manuscript.
>
> **Choice of $p_{\text{critical}}$.** Indeed, the choice of $\sigma_t$ as the critical probability is a heuristic, and specific only to DDPM-based diffusion models. We do note that most other inverse algorithms we compare against (DPS, Latent-DPS, MCG, DDNM, DDRM, PSLD, STSL, ReSample) are generally derived for VPSDE sampling, and not VESDE sampling. We agree that this is a limiting design, and that this can be an important direction for future work.

---

> ### Comment · Reviewer_bebv · 2024-11-22
> **Response**
>
> **On Tweedie’s formula**
>
> It is true that Tweedie’s formula is used as a proxy for $x_0$. However, this does not justify misrepresenting it.
> Equation (17) remains unclear.
> Given access to the noisy version $x_t$, Tweedie’s formula provides $E(x_0 | x_t)$, not $x_0$ itself.
> Moreover, the claim in Line 295-297 that states *"using Tweedie’s formula requires $p_t(x_t|x_0)$ to be normally distributed, while the data distribution $p_{\text{data}}$ modeled by the diffusion model is usually highly multimodal"* is incorrect.
> Tweedie’s formula holds in diffusion models because the transition kernels are Gaussian.
>
> For a rigorous explanation, refer to Section 2.3 of [1] and Proposition 1 in [2].
>
> **Comparison with latent space models**
>
> The comparison between pixel-space and latent-space models is misplaced, as the following two problems are fundamentally different:
>
> 1.
>    $
>    \min_x \|| y - A(x) \||^2 \quad \text{s.t.} \quad x \sim p_{\text{pixel}}(x)
>    $
> 2.
>    $
>    \min_x \|| y - A(x) \||^2 \quad \text{s.t.} \quad x \sim p_{\text{latent}}(x)
>    $
>
> It is therefore irrelevant to compare an algorithm solving Problem (1) with one solving Problem (2).
> A standard approach would be to compare algorithms on Problem (1) and separately on Problem (2), rather than running some Problem (1) and others on Problem (2).
>
> **Fundamental Mistakes**
>
> Despite revisions, the paper still contains serious theoretical errors:
>
> - In Equation (9) does not correspond to DDPM [3] sampling scheme
> - The treatment of Tweedie’s formula is problematic, see the first point **On Tweedie’s Formula**
> - In Line 306, the claim in Line 305-306 *"While at $t \approx 0$, $p_t(x_t)$ may approach an isotropic Gaussian"* is incorrect.
> At, $t \approx 0$, $p_t(x_t)$ closely resembles $p_0(x_0)$ the data distribution and is highly multimodal.
> - In Equation (17), It should be $p_t(x_t)$, not $p_t(x_t | x_0)$. Additionally, the left-hand side is missing an expectation operator.
> - In Line 5 of Algorithm 2, the gradient should be taken with respect to $\log p_t$ not $p_t$
>
>
> **Final Notes**
>
> The authors aim to provide an estimator for $E(X_0 | X_t, y)$, a generally intractable quantity. However, this objective is never explicitly stated in the paper. Instead, it is motivated by a misleading treatement of Tweedie’s formula.
>
>
> ---
>
> .. [1] Meng, Chenlin, et al. "Estimating high order gradients of the data distribution by denoising." Advances in Neural Information Processing Systems 34 (2021): 25359-25369.
>
> .. [2] Chung, H., Kim, J., Mccann, M. T., Klasky, M. L., & Ye, J. C. (2022). Diffusion posterior sampling for general noisy inverse problems. arXiv preprint arXiv:2209.14687
>
> .. [3] Ho, Jonathan, Ajay Jain, and Pieter Abbeel. "Denoising diffusion probabilistic models." Advances in neural information processing systems 33 (2020): 6840-6851.

---

> ### Author Response · Authors · 2024-11-22
>
> We thank the reviewer for their further thoughts on our submission. We address their concerns in a detailed response below.
>
> **On Tweedie's formula**
>
> We respond to the reviewer point-by-point here.
>
> *Misrepresenting Tweedie's formula.* We apologize for the initial confusion in our original submission surrounding Tweedie's formula. However, we have fixed that. We do not characterize Eq. 17 as Tweedie's formula anywhere in the current text. Note that Eq. 17 is a formula for predicting $x_0$ **based** on Tweedie's, but **holds due to the proof of Lemma 3.1, irrespective of Tweedie's.** We firmly maintain its correctness.
>
> In general, we had adjusted any mention of Tweedie’s accordingly in the last revision, and include a explicit description of our approach (namely $E(x_0|x_t)$ vs $x_0$ prediction) before Eq. 4 (Line 75-76).
>
> *Claim in 295-297.* We thank the reviewer for catching this typo. We meant to write $x_t$ here. We have made this change and apologize for the confusion.
>
> **Comparison with latent space models**
>
> Indeed, this is a possible characterization of the problem. However, we would argue that $p_\text{pixel}$ and $p_\text{latent}$ are **both approximations of $p_\text{data}$**, and therefore both 1. and 2. are surrogates for the underlying problem
>
> 3. $ \min_x || y - A(x) ||^2 \quad \text{s.t.} \quad x \sim p_\text{data}(x). $
>
> We can see the reviewer's point that using $p_\text{pixel}$ and $p_\text{latent}$ to approximate $p_\text{data}$ may constitute a different set of challenges to overcome. However, in the literature on diffusion-based solvers for inverse problems, it is **standard to make such a comparison**, see [1, 2, 3, 4]. Moreover, we believe that a comprehensive comparison may be useful for practitioners and researchers who are interested in solving 3. at large.
>
> To further distinguish between latent and pixel space models, we have added a dividing line between the two solver types in Tables 1, 3 and 4 and the additional context "We compare against pixel-based solvers (upper half) and latent-based solvers (lower half)." in the table captions.
>
> **Fundamental mistakes**
>
> We thank the reviewer for pointing out these misunderstandings due to typographical mistakes.
>
> - Eq. 9: We have corrected this typo from DDPM -> DDIM with $\sigma_t = \sqrt{1 - \alpha_t}$.
> - Tweedie's formula treatment: In the previous revision, we had corrected all references to Tweedie's formula to "predictions of $x_0$ with Tweedie's formula". *We outline the exact mechanism of this Tweedie's based prediction in Eq. 4.* For a discussion on Eq. 17 see our reply to **On Tweedie's formula**.
> - Line 306: We have corrected this typo. $p_t(x_t)$ -> $p(x_0|x_t)$.
> - Eq. 17: As discussed in our reply to **On Tweedie's formula**, we do not characterize Eq. 17 as Tweedie's formula. Eq. 17 is correct due to the proof of Lemma 3.1 containing Eq. 17.
> - Line 5 of Algorithm 2. We have corrected this typo.
>
> **Final notes**
>
> *Estimating $E(x_0 | x_t, y)$.* Indeed, this is one possible characterization of our algorithm which we considered. We note that this characterization is non-specific, since it can also apply in general to the $\hat{x}_0$ estimate in the reverse diffusion step of any ($y$-, text-, class-, etc.) conditional diffusion model. However, we appreciate the connection and have added a statement of this characterization to Section 1 for a more comprehensive discussion.
>
> *Misleading treatment of Tweedie's.* We hope that our discussion in the dedicated section above clarifies any remaining confusion.
>
> [1] Solving Inverse Problems with Latent Diffusion Models via Hard Data Consistency. https://arxiv.org/abs/2307.08123
>
> [2] Solving Linear Inverse Problems Provably via Posterior Sampling with Latent Diffusion Models. https://arxiv.org/abs/2307.00619
>
> [3] Beyond First-Order Tweedie: Solving Inverse Problems using Latent Diffusion. https://arxiv.org/abs/2312.00852
>
> [4] Prompt-tuning latent diffusion models for inverse problems. https://arxiv.org/abs/2310.01110

---

### Official Review · Reviewer_M6gj · 2024-11-04

**Soundness:** 1
**Presentation:** 3
**Contribution:** 2
**Rating:** 3
**Confidence:** 4

**Summary:**

This paper proposed a diffusion posterior sampling method which made a correction to the score function by a Maximum Likelihood Estimation (MLE). While the MLE is usually ill-defined since in many cases the measurement operator is underdetermined, this paper proposed a remedy by running gradient descent from a good initialization and introducing an early stopping criterion to compute MLE.

**Strengths:**

The proposed algorithm is simple and has a better performance than DPS.

**Weaknesses:**

1. The experimental results are confusing. It is possible that some of the baselines were not implemented correctly.
- In the experiments, many of the more recent algorithms, including PSLD, ReSample, had worse performance than DPS in most settings. This is apparently different than what was reported in the literature. I also have some personal experience of implementing these algorithms, and they all demonstrated clear advantage over DPS in my setting.
- The paper used Stable Diffusion v1.5 for some of the baselines while using a specialized ImageNet score network for the proposed algorithm. This is clearly unfair.
- Appendix C.4 seems to have some serious concerns about ReSample. However, not enough evidence was provided to support their arguments.

2. While one of the major advantages of the paper was claimed to be less computation as it did not require computing Jacobian, this did not take into account possible variants of previous algorithms like DPS, LGD-MC. There are easy ways to get rid of Jacobian in these algorithms, and there should be thorough comparison with them.

**Questions:**

Please refer to the Weaknesses part.

---

> ### Author Response · Authors · 2024-11-17
>
> We thank the reviewer for their helpful discussion and insightful critique. We hope to clarify some of the points of the paper, and alleviate the reviewer's concerns below in a point by point response.
>
> **It is possible some baselines were not implemented correctly.** We take the reviewer’s concerns very seriously. We firmly believe that we have properly implemented the baselines, and invite the reviewer to verify this with the provided code in Footnote 2 (https://anonymous.4open.science/r/diffusion_conditional_sampling/README.md).
>
> **In the experiments, PSLD and ReSample had worse performance than DPS in most settings.** Indeed, there is a discrepancy between our evaluations and the self-reported performance of PSLD and ReSample. This is due to several reasons, which we covered in the Appendix of the original submission but also discuss below.
>
> Generally, we found that both perform well in the original low / zero noise settings of their respective studies, but underperform compared to DPS at the higher noise levels in our work due to their noise-sensitive inner optimization loop, which DCS ameliorates with the proposed noise-aware optimization.
>
> Moreover, the underperformance of PSLD (w.r.t. self-reported values) has been corroborated in other studies, such as [1] and [2].
>
> Additionally, ReSample was originally evaluated on a heavily downsampled subset of the full FFHQ-1K dataset that was not made public (see Section 4.1 in their paper [1]). Most works in this field such as DPS, DDNM, PSLD, MCG, consider the full test set of FFHQ-1K. On this full set, ReSample performance differs greatly from this reported result. Due to this discrepancy, we also provided a 100 image subset of FFHQ-1K in the original submission that recovers the reported performance of ReSample (Table 4), where we still demonstrate significant improvements of DCS over ReSample.
>
> **The paper uses a specialized ImageNet network while other baselines use Stable Diffusion v1.5.**
> Note that the majority of baselines we compare against (e.g., DPS, MCG, DDNM, DDRM) use the same specialized ImageNet score network that we use, and we obtain state-of-the-art results among these baselines.
>
> Only latent baselines use Stable Diffusion, due to the absence of a similarly specialized latent diffusion model. It is possible to train a separate latent diffusion model solely on ImageNet images for this comparison, but as an academic lab, we did not have sufficient resources to do so. Moreover, we note that comparisons between latent and pixel-space models in this manner is an established practice [3, 4], and SD v1.5 makes up for the non-specialized training with its more than 2.5x times more parameters than the ImageNet model. Therefore, we argue that this comparison still provides a meaningful data point, and is preferable to omitting latent models altogether for this comparison.
>
> **One of the major advantages of DCS is that it requires less computation, but the paper does not take into account Jacobian-free variants of DPS and LGD-MC.**
> We thank the reviewer for this suggestion. We have added a thorough evaluation of Jacobian-free DPS (DPS-JF) and Jacobian-free LGD-MC (LGD-MC-JF) to our experiments. We ran a full suite of experiments on FFHQ-1K in the updated Table 1.  For LGD-MC-JF we set the number of Monte-Carlo samples to 10. Our conclusions are summarized below.
>
> Overall, the Jacobian-free variants of DPS and LGD-MC are significantly inferior in sample quality to DCS, especially when constrained to the same computational budget as DCS (see $T=100$ variants).
>
> Even at a higher computational budget where we use the same number of timesteps as DPS, DPS-JF and LGD-MC-JF appear to suffer from a significant drop in performance compared to DPS, which is already inferior to DCS. At this setting, DPS-JF is slower than DCS (by a factor of 1.5x).
>
> Ultimately, the optimal implementation of DPS-JF is still considerably worse than DCS, since it requires many more diffusion steps $T$ to saturate in performance. Conversely, DCS benefits from the empirically fast convergence of NAM and is very robust to $T$, which we are able to set much lower than other algorithms (see updated Figure 7).
>
>
> [1] Resample. https://arxiv.org/abs/2307.08123
>
> [2] Decoupled Data Consistency with Diffusion Purification for Image Restoration. https://arxiv.org/abs/2403.06054
>
> [3] Beyond First-Order Tweedie: Solving Inverse Problems using Latent Diffusion. https://arxiv.org/abs/2312.00852
>
> [4] Prompt-tuning latent diffusion models for inverse problems. https://arxiv.org/pdf/2310.01110

---

> > ### Author Response · Authors · 2024-11-20
> >
> > **Appendix C.4 seems to have some serious concerns about ReSample. However, not enough evidence was provided to support their arguments.**
> >
> > To reiterate, there are **two** notable aspects of ReSample we highlight in Appendix C.4. We would not consider them concerns, but they affect the interpretation and empirical validation of the algorithm.
> >
> > **First**, their GitHub implementation differs from their proposed Algorithm 1. Namely, they claim to run only an unconditional DDIM step in addition to their consistency optimization. However, they add a DPS update to each sampling step [here](https://github.com/soominkwon/resample/blob/03f5d069953cad42f8e0f8f44cddb6bed375ce91/ldm/models/diffusion/ddim.py#L255). `measurement_cond_fn` is the class method `conditioning` of `PosteriorSampler` defined [here](https://github.com/soominkwon/resample/blob/03f5d069953cad42f8e0f8f44cddb6bed375ce91/ldm_inverse/condition_methods.py#L58), which contains
> > ```
> > norm_grad, norm = self.grad_and_value(x_prev=x_prev, x_0_hat=x_0_hat, measurement=measurement, **kwargs)
> > x_t -= norm_grad * scale
> > ```
> > This computes Line 7 of Algorithm 1 in DPS
> > $x_t ‎ = x_t - \zeta_t \nabla_{x_t} || y - A(\hat{x}_0)||_2^2$, which is the neural network Jacobian-based update in DPS. This greatly increases GPU memory usage.
> >
> > **Second**, the runtime of ReSample is slow. With an RTX A6000 the super-resolution task on FFHQ takes ~8 min/img. (c.f. our algorithm which takes <0.5 min/img or DPS which takes ~2 min/img.) This can be easily verified by running our implementation, or their original code [here](https://github.com/soominkwon/resample).
> >
> > Finally, in Appendix B, we exactly replicate the settings in the ReSample paper on the FFHQ-1K dataset in Table 4, reproducing the results from their paper. Here, we find that DCS still outperforms ReSample. Tables 1 and 3 further illustrate how ReSample’s noise robustness is poor compared to DCS.

---

### Meta-Review · Area_Chair_Muuy · 2024-12-12

**Metareview:**

The paper proposed a new method for solving linear inverse problems with diffusion priors. As the proposed algorithm is heuristic, its success is mainly supported by extensive numerical experiments. However, there are many outstanding concerns and doubts regarding the numerical experiments in terms of the baselines and comparisons; for example, separate comparisons should be performed for the latent-space version and the pixel-space version, inconsistent results compared to what have been reported in the literature, etc. Therefore, the paper is recommended for rejection in this round.

**Additional Comments On Reviewer Discussion:**

The discussion has been centered around the technical correctness and the experiments, and while making some progress, the majority of the reviewers still hold reservation about this work.

---

### Decision · Program_Chairs · 2025-01-22

Reject